# Assessing the role of thermal disequilibrium in the evolution of the lithosphere-asthenosphere boundary: An idealized model of heat exchange during channelized melt-transport

Mousumi Roy[1]

[1]Department of Physics and Astronomy, University of New Mexico, Albuquerque, NM 87106, USA

**Correspondence:** Mousumi Roy (mroy@unm.edu)

**Abstract.** This study explores how the continental lithospheric mantle (CLM) may be heated during channelized melt transport when there is thermal disequilibrium between (melt-rich) channels and surrounding (melt-poor) regions. Specifically, I explore the role of disequilibrium heat exchange in weakening and destabilizing the lithosphere from beneath as melts infiltrate into the lithosphere-asthenosphere boundary (LAB) in intraplate continental settings. During equilibration, hotter-than-ambient melts would be expected to heat the surrounding CLM, but we lack an understanding of the expected spatio-temporal scales and how these depend on channel geometries, infiltration duration, and transport rates. This study assesses the role of heat exchange between migrating material in melt-rich channels and their surroundings in the limit where advective effects are larger than diffusive heat transfer (Péclet numbers > 10). I utilize a 1D advection-diffusion model that includes thermal exchange between melt-rich channels and the surrounding melt-poor region, parameterized by the volume fraction of channels ($\phi$), average relative velocity ($v_{channel}$) between material in/out of channels, channel spacing ($d$), and timescale of episodic/repeated melt-infiltration ($\tau$). The results suggest that: (1) during episodic infiltration of hotter-than-ambient melt, a steady-state thermal reworking zone (TRZ) associated with spatio-temporally varying disequilibrium heat exchange forms at the LAB. (2) The TRZ grows by the transient migration of a disequilibrium-heating front at a material-dependent velocity, reaching a maximum steady-state width $\delta$ proportional to $[\phi v_{channel}(\tau/d)^n]$, where $n \approx 2$ for periodic thermal perturbations and $n \approx 1$ for a single finite-duration thermal pulse. For geologically-reasonable model parameters, the spatio-temporal scales associated with establishment of the TRZ are comparable with those inferred for the migration of the LAB based on geologic observations within continental intra-plate settings, such as the western US. The results of this study suggest that, for channelized transport speeds of $v_{channel} = 1$ m/yr, channel spacings $d \approx 10^2$ m, and timescales of episodic melt-infiltration $\tau \approx 10^1$ Kyr, the steady-state width of the TRZ in the lowermost CLM is $\approx 10$ km. (3) Within the TRZ, disequilibrium heat exchange may contribute $\approx 10^{-5}$ W/m$^3$ to the LAB heat budget.

## 1 Scientific Motivation

During its long residence at Earth's surface, continental lithosphere is shaped by tectonic events such as rifting (including supercontinent break-up) and plate collision, undergoing profound changes in its physical and chemical state. In some cases, previously stable (undeforming) portions of the continental lithosphere may be destabilized. Based on the close association

of magma infiltration with these events there is growing speculation that, under certain circumstances, melt-rock interaction may somehow weaken and perturb the stability of continental lithosphere (Hopper et al., 2020; Wenker and Beaumont, 2017; Plank and Forsyth, 2016; Roy et al., 2016; Wang et al., 2015; Menzies et al., 2007; Carlson et al., 2004; Gao et al., 2002; O'Reilly et al., 2001). Recent work in subduction settings suggests that heat advection by magma transport into the overriding lithosphere is a fundamental process that determines the thermal structure of arcs and possibly aids thinning of the overriding

plate (England and Katz, 2010; Perrin et al., 2016; ReesJones et al., 2018).

Fundamentally, this melt-enhanced weakening of continents is intertwined with the notion of thermal and chemical disequilibrium between infiltrating melts and the surrounding material, and therefore with *transient* processes that drive the lithosphere from one stable state to another. These processes, however, remain elusive. This paper explores one aspect of the problem, namely, thermal disequilibrium during infiltration of hotter-than-ambient melts into the base of the lithosphere, as a means of

35 weakening and shaping the continental lithosphere from beneath. Specifically, I explore thermal disequilibrium between melt-rich channels and surrounding (melt-poor) material as a process to heat and modify the continental lithospheric mantle (CLM). The models explored here are a useful way to place constraints on the melt-transport scenarios for which a significant degree of disequilibrium heat exchange (driven by temperature differences between materials within and surrounding channels) may be important in the CLM at or near the lithosphere-asthenosphere boundary (LAB).

This study is inspired by evidence for the role of thermal disequilibrium from detailed field-based, petrologic, and geochemical studies in the Lherz and Ronda peridotite massifs (Bodinier et al., 2008, 1990; LeRoux et al., 2007; Lenoir et al., 2001; Saal et al., 2001; Vauchez and Garrido, 2001; LeRoux et al., 2008, 2007, 2008; Soustelle et al., 2009). Important conclusions from studies in the Lherz and Ronda peridotite massifs include: (1) "lherzolite" (named after its type-section in the Lherz massif), commonly regarded as pristine, fertile sub-continental lithospheric mantle, may be derived from refertilization of a

depleted, harzburigitic parent (e.g., LeRoux et al., 2007, 2008); (2) Careful microstructural, geochemical and petrologic work has documented the dominant effect of a steep thermal gradient associated with the region of contact and interaction between partial-melt-rich regions and the surrounding lithosphere. These workers provide a quantitative estimate of a transient thermal gradient ($\approx 230^{\circ}$C/km, or more than an order of magnitude larger than a typical equilibrium geothermal gradient expected at the LAB). Indeed, the authors recognize this as a transient LAB and coin the term "asthenospherization" for disequilibrium

heat exchange processes modifying the LAB. The spatial scale over which this disequilibrium heating is observed in Ronda ($\sim 1$ km) guides the mesoscale modeling approach here.

Additionally, this work is motivated by observations from the western US, which has undergone extensive magma-infiltration in Cenozoic time. Pressures and temperatures of last equilibration of Cenozoic basalts consistently point to depths that are at or below the present LAB (Plank and Forsyth, 2016), suggesting that melt transport from those depths upward through the lower

CLM occurs in thermal disequilibrium. In the Big Pine volcanic field, for example, the inferred depth of the LAB decreases by >10 km in a timespan of <1 Myr (Plank and Forsyth, 2016), suggesting that the processes associated with this migration may be transient (LAB vertical migration rates in excess of $\approx 10$ km/Myr). More recently, Cenozoic melt- or fluid-enhanced thinning of the CLM in the western US has also been inferred from geochemical and isotopic data from volcanic rocks (Farmer et al., 2020). Such processes of thermally-driven erosion/migration of the LAB are also inferred in arc settings (e.g., Perrin

et al., 2016; ReesJones et al., 2018). Motivated by these observations, a primary goal of this work is to quantify the role of transient, disequilibrium heating by infiltrating channelized melt as a mechanism for modifying the LAB and the lowermost CLM.

It has been argued that a permeability contrast (e.g., Holtzman and Kendall, 2010) or a change in magma mobility (e.g., Sakamaki et al., 2013) across the LAB is likely to drive melts to pond and possibly drive the upward propagation of dikes that may freeze and heat the CLM (e.g., Havlin et al., 2013; ReesJones et al., 2018). Not all infiltrating melts would freeze, however, and some component of hotter-than-ambient melts may be transported in thermal and chemical disequilibrium into the CLM via established channels/pathways (e.g., LeRoux et al., 2007; Schmeling et al., 2018). Thermal disequilibrium during melt transport is expected to become important within the CLM as the degree of channelization and the relative melt-solid velocity increases (e.g., Schmeling et al., 2018; Chevalier and Schmeling, 2022). In this work, I am not concerned with the emergence and development of these channel networks within the lower CLM, nor the deeper processes that transport melt from a sub-lithospheric melt-generation zone (e.g., Aharonov et al., 1995). Instead, the starting point of this study is the observation that high-porosity, melt-rich channels are an important part of melt-rock interaction in the CLM (e.g. Soustelle et al., 2009; LeRoux et al., 2007). An important limitation of the models, therefore, is that channelization is imposed via parameters that control a heat transfer coefficient. The simplicity of the models, however, allows us to focus on the implications of significant thermal gradients between melt-rich channels and their surroundings. Although others have also argued for the important role of thermal disequilibrium in melt-rock interaction (Chevalier and Schmeling, 2022; Keller and Suckale, 2019; Wallner and Schmeling, 2016; Schmeling et al., 2018), this study provides a quantification of the role of thermal disequilibrium at the LAB based on observational constraints discussed above. This work builds on the 1D model in Roy (2020) (which did not consider axial conduction) and includes both a thermal contrast that drives heat exchange and axial diffusion terms (e.g., following Chevalier and Schmeling, 2022) in order to explore thermal equilibration over long timescales of melt infiltration ($> 10^{3-4}$ yrs) into the lowermost 1-10 km of the CLM (e.g., stage 3, large Péclet numbers in Chevalier and Schmeling, 2022).

The 1D model abstracts the complex geometry of the melt-rock interface and therefore differs from other descriptions of disequilibrium heat exchange (Wallner and Schmeling, 2016; Schmeling et al., 2018; Keller and Suckale, 2019). Similar to chemical transport models that use a linear driving term for chemical disequilibrium (e.g., Hauri, 1997; Kenyon, 1990; Bo et al., 2018), the models below assume a linear thermal driving term (e.g. Schumann, 1929; Kuznetsov, 1994; Spiga and Spiga, 1981). In this study, however, only the role of thermal disequilibrium is included and I ignore important processes such as chemical disequilibrium (e.g., Hauri, 1997; Kenyon, 1990; Bo et al., 2018). The basic results of the 1D model are presented below, followed by further discussion of their limitations and implications. Although idealized, the models provide a first-order assessment of the temporal and spatial scales over which thermal disequilibrium can play a role in warming and therefore weakening the lowermost CLM. A key finding is that the lowermost portion of the CLM may evolve into a long-term thermal reworking zone (TRZ) driven by disequilibrium heat exchange in channelized melt transport. The factors that determine the rate at which the TRZ is formed and its spatial scale are estimated from the models and are compared to geologic observations within the western US, specifically geochemical and petrologic evidence for the upward migration of the LAB during Cenozoic melt-rock interaction (Plank and Forsyth, 2016; Farmer et al., 2020).

## 2  Model of disequilibrium heat transport

The starting point for the models explored here is a simple, 1-D theory of heat exchange in packed porous beds by Schumann (1929), building on Roy (2020), where fluid moves within the pores of a matrix of solid grains and the only variability in temperature (and temperature contrast between solid and fluid) is in the transport direction. In Schumann (1929), the thermal evolution of the system is governed mainly by heat exchange across the solid-fluid interfacial surface. In this study, I present a re-interpretation of Schumann's model and of the effective heat transfer coefficient. Instead of considering fluid moving in pores between solid grains, the system of equations Schumann (1929) may be used to describe thermal disequilibrium between material within high-porosity channels and outside channels. In other words, here "fluid" is interpreted to be in-channel material and "solid" is material outside channels (for simplicity, however, I retain the subscripts $f$ (in-channel) and $s$ (outside channels)). The goal here is to describe the relative importance of advective transport over length scales that are comparable to the channel spacings, so we define a transport velocity, $v_{channel}$, as an average relative velocity across channel walls. As described below, this reinterpretation also extends to the physical meaning of the effective heat transfer coefficient, where now the geometry across which the transfer occurs must take into account the channel geometry and spacing. Furthermore, following Schumann (1929), heat exchange is assumed to be linearly proportional to the local temperature difference between solid and fluid (see also Roy (2020)). Unlike Schumann (1929) and Roy (2020), however, this study includes thermal diffusive effects to account for (axial) conduction within channels and within the material outside of the channels. These arguments lead to coupled equations for the temperature outside channels, $T_s$, and within channels, $T_f$:

$$\frac{\partial T_f}{\partial t} + v_{channel}\frac{\partial T_f}{\partial x} = -\frac{K}{\phi c_f}(T_f - T_s) + \frac{\lambda_f}{c_f}\frac{\partial^2 T_f}{\partial x^2} \equiv -K_f(T_f - T_s) + \frac{\lambda_f}{c_f}\frac{\partial^2 T_f}{\partial x^2} \tag{1}$$

$$\frac{\partial T_s}{\partial t} = \frac{K}{(1-\phi)c_s}(T_f - T_s) + \frac{\lambda_s}{c_s}\frac{\partial^2 T_s}{\partial x^2} \equiv K_s(T_f - T_s) + \frac{\lambda_s}{c_s}\frac{\partial^2 T_s}{\partial x^2} \tag{2}$$

where $v_{channel}$ is the average transport velocity of material within melt-rich channels relative to the melt-poor surrounding material, $\phi$ is a fluid volume fraction, $\lambda_f$ and $\lambda_s$ are thermal conductivities, and $c_f$ and $c_s$ are the heat capacities per unit volume (heat capacitances) at constant pressure ($c_f = c_{pf}\rho_f$ and $c_s = c_{ps}\rho_s$), and $x$ is the position coordinate in the transport direction. Note that the geometry of the solid-fluid interface is not treated in detail, but is idealized in the channel volume fraction, $\phi$, the channel spacing, $d$, both of which control an effective (across-channel-wall) heat transfer coefficient, $K$, discussed below.

One advantage of "coarse-graining" the Schumann (1929) model (from the pore scale to macroscopic channels) is its simplicity and that it has been investigated in numerous previous studies. There exist analytic solutions for Eqns 1 and 2 in limiting cases, particularly for large Péclet number with axial diffusion terms ignored (without the last terms in Eqns 1 and 2) (Spiga and Spiga, 1981; Kuznetsov, 1994, 1995a, b, 1996). This re-interpretation of the model must also be accompanied by an appropriate reinterpretation of the heat transfer coefficient, $K$, made possible because in the framework above the geometry of the interfacial surface is not explicitly specified. The reinterpreted model is applied to a semi-infinite domain where fluid transport

occurs in high-porosity channels aligned in one direction (Figure 1). The high-porosity channels are assumed to occupy a constant volume fraction, $\phi$, within which material moves with a constant (average) velocity $v_{channel}$ relative to the surrounding stationary material outside the channel (volume fraction $1-\phi$). The model domain may be thought of as co-moving with the reference frame of material outside the channels. Because of the assumptions built-in to the 1D approach, the results below are applicable to physical situations where transport is dominantly in the along-channel direction and any motion of material outside channels is steady. The model assumes that the average channel geometry is unchanging within the domain and that the material outside the channels is initially at a uniform temperature. The models and their interpretations, therefore, are used here to assess the role of thermal disequilibrium as melts are transported ($\approx 10-20$ km or so) within the lowermost CLM above the LAB within intraplate settings.

In this model, the terms involving the thermal contrast ($T_f - T_s$) in Eqns 1 and 2 represent heat transfer across the walls of the channels, between material within and outside the channels. This heat exchange depends on material parameters that govern thermal diffusion perpendicular to the transport direction, and on the geometry of the channels themselves (e.g., sinuosity, spacing, etc.). The detailed geometry of the channel walls (the relevant interfacial surface here) is not specified in this simple 1D treatment, but is parameterized by the heat transfer coefficient, $K$ (Figure 1). Therefore, $K$ is a proxy for the efficiency of conduction perpendicular to the transport direction, and the geometry of the channel wall interface, namely the wall area per unit volume, controlled by the spatial scale of channelization, $d$ (see 2.1). As illustrated in Figure 1, a large value of $K$ may represent efficient heat exchange as in the case of many channels separated by a small distance. Conversely, a low value of $K$ would represent inefficient exchange, as in the case of a larger characteristic separation between the channels. In the following sections, I consider the physical meaning of $K$, and also present a non-dimensionalization of the Eqns 1 and 2 based on characteristic length and timescales in the problem. The coefficients of the temperature-contrast terms on the right hand sides of Eqns 1 and 2 specify the timescales of heat exchange within channels, $t_f = 1/K_f = \phi c_f / K$, and outside channels, $t_s = 1/K_s = (1-\phi)c_s/K$. Additionally, a characteristic length scale emerges out of the relative motion across channel walls, $v_{channel}/K_f$. These characteristic length and timescales are used to non-dimensionalize Eqns 1 and 2 (see 2.2) and obtain the results presented below.

## 2.1 Heat transfer coefficient

In this section I consider the meaning of the heat transfer coefficient, $K$, and the related constants, $K_s = K/(c_s(1-\phi))$ and $K_f = K/(c_f\phi)$ in Eqns 1 and 2. Note that $K_f$ and $K_s$ have dimensions of inverse time, and they both depend on the heat transfer coefficient $K$. Physically, $K$ represents the amount of heat transferred across channel walls per unit time, per unit volume, per unit difference in temperature (in Schumann, 1929, this exchange is across the solid-fluid interface). The factors that determine $K$ can be illustrated by considering that the heat transfer rate across channel walls must depend on the geometry of walls and also on the effective thermal conductivity of the channelized domain.

Although the geometry of the channels may be complex, I consider one aspect of it, namely, the specific wall surface area (wall area per unit volume), $a_{sf}$, which is a function of the spatial scale of channelization. In the grain-scale porous flow case considered in Schumann (1929) for example, if the solid matrix is made of spheres with an average particle diameter $p$, then

the specific area for a grain is $S_0 = 6/p$, so $a_{sf} = S_0(1 - \phi) = 6(1 - \phi)/p$ (Dullien, 1979). This sets a limit for channels with channel spacing $d$, where we shall assume that the specific surface area is $a_{sf} \approx A(1 - \phi)/d$, where $A$ is a number that is between 2 (for planar sheet-like channels with small volume fraction $\phi$) and 6 for transport around spherical regions (Dixon and Cresswell, 1979; Schmeling et al., 2018). Whereas the specific wall area is a geometric factor, the effective conductivity of the medium depends on the Nusselt number, $Nu$. Theoretical arguments in Dixon and Cresswell (1979) show that the effective thermal conductivity may be written in terms of the individual in-channel and out-of-channel thermal conductivities $\lambda_f$ and $\lambda_s$ (equivalent to considering the channels and non-channel regions in parallel):

$$\frac{1}{C_{eff}} = \left[ \frac{1}{Nu\lambda_f} + \frac{1}{\beta\lambda_s} \right] \tag{3}$$

where $\beta = 10$ for spherical matrix grains, 8 for cylinders, and 6 for slabs (Dixon and Cresswell, 1979). Therefore, the range of $\beta = 6$ to 10 represents the highly-channelized vs porous flow end-members. For slow flows (Reynolds number $Re \ll 100$), Handley and Heggs (1968) argue that $Nu$ ranges from $0.1$ to $12.4$ (Dixon and Cresswell, 1979) (Table 1). The relevant quantity that determines $K$ is an effective "conductance" $C_{eff}/d$, so that $K = C_{eff}a_{sf}/d$,

$$K = \left[ \frac{1}{Nu\lambda_f} + \frac{1}{\beta\lambda_s} \right]^{-1} \frac{A(1 - \phi)}{d^2} \tag{4}$$

a product of a material-dependent quantity and a geometry-dependent quantity.

Turning now to physical properties relevant to the transport of melts through the lithosphere, the channelized domain may be thought of as consisting of a mixture of melt+grains throughout, but with variable grain-scale porosity $\varphi$. Specifically, the fluid-rich channels would have a higher $\varphi_f$ than the surroundings $\varphi_s$. The thermal conductivity inside and outside the channels would then be a volume average in each, e.g., inside channels, $\lambda_f = \varphi_f\lambda_{melt} + (1 - \varphi_f)\lambda_{grain}$, where $\lambda_{melt}$ and $\lambda_{grain}$ are the values for, say basaltic melt and peridotitic grains. Similarly, outside channels the volume average would be $\lambda_s = \varphi_s\lambda_{melt} + (1 - \varphi_s)\lambda_{grain}$. Here I do not specify reasonable ranges of $\varphi_f$ and $\varphi_s$ (in general, they will both be within $[0, 1]$), but rather I focus on determining an upper limit to the role of disequilibrium heat exchange across channel walls. Therefore, to explore the end-member case, I take $\varphi_f = 1$ and $\varphi_s = 0$, so that $\lambda_f = \lambda_{melt}$ and $\lambda_s = \lambda_{grain}$. Using a reasonable conductivity for basaltic magma of $\lambda_{melt} = 1$ W m$^{-1}$ K$^{-1}$ (Lesher and Spera, 2015), $\lambda_{grain} = 2.5$ W m$^{-1}$ K$^{-1}$) for the solid grains, and taking $Nu = 0.1$ to $12.4$, $\beta = 6$, 8, or 10, and $A = 6$, we find that the effective conductivity $C_{eff}$, $a_{sf}$ and $K$ are within the ranges shown in Figure 2. (Note that choosing $A \approx 2$ would change $K$ by less than an order of magnitude.) As suggested by Eqn (4), $K \sim d^{-2}$ and strongly decreases with increasing spatial scale of channels; Figure 2c. Although Figure 2 theoretically explores the full range of parameters and their effect on $K$, in practice these parameters are not independent and should be chosen based on the geometry considered (e.g., Chevalier and Schmeling, 2022). In the models in section 3 I set $\beta = 6$ and $A = 2$ as suggested for 1D channels (Dixon and Cresswell, 1979; Schmeling et al., 2018). As discussed in Schmeling et al. (2018), the heat transfer coefficient should depend not only on $d$ the channel spacing, but also a length scale set by the thickness of a microscopic thermal boundary layer at fluid-solid interfaces. Specifically, $K \sim C_{eff}/$(a boundary

layer dimension) (Schmeling et al., 2018). This boundary layer thickness is a function of time and only at timescales that are long relative to a characteristic thermal response time will the boundary layer encompass the entire region between channels Schmeling et al. (2018). Therefore, by taking $K \sim d^{-2}$, our models must be confined to timescales that are large relative to the material-dependent thermal response timescale (e.g., $1/K_s$), a requirement that is met in all thermal perturbations considered in this study.

To decide on a range of $K$ values appropriate to the LAB, I turn to geologic observations of the scale of channelization in exhumed portions of the lower CLM. Structural, petrologic, and geochemical data from the Lherz Massif suggest that melt-rock interaction has driven refertilization of a harzburgite body into lherzolite (LeRoux et al., 2007, 2008). In the field, the lherzolite bodies are separated from each other by distances of several tens of meters and this is also the spatial scale of isotopic disequilibrium between metasomatizing fluids and the harzburgite parent material (LeRoux et al., 2008). With this as a proxy for the spatial separation of fluid-rich channels, I choose a broad range for the relevant spatial scale of channelization, $d = 10^0$ to $10^3$ m (1 m to 1 km channel spacing). The corresponding range of the heat transfer coefficient in the models is therefore $K \approx 10^{-5}$ (large $d$) to $10^1$ (low $d$) in W m$^{-3}$K$^{-1}$ (Figure 2). In the following, material properties, channel volume fraction, channel velocity, and heat transfer coefficient are fixed for each calculation (Table 1), but we confine $d > 10^2$ m in order to ensure an effective Péclet number $Pe > 10$. Taking typical parameters $\phi = 0.1$, $d = 1000$ m, and corresponding $K$ values in Figure 2, typical timescales of response are $t_s = 1/K_s \approx 9800$ yrs and $t_f = 1/K_f \approx 1200$ yrs, short compared to the timescales of geologic events and the timescale of sinusoidal and pulse-like thermal perturbations considered here.

## 2.2 Nondimensional system

To non-dimensionalize the system of equations 1 and 2 (following Spiga and Spiga, 1981), we define the normalized relative temperature, $T'_f = (T_f - T_0)/\Delta T$ and $T'_s = (T_s - T_0)/\Delta T$, where $T_0$ is reference temperature and $\Delta T$ is a temperature perturbation (described below). We also introduce the dimensionless position, $x' = xK_f/v_{channel}$, a dimensionless time, $t' = K_s t$, and the weighted heat capacitance ratio, $\zeta = K_s/K_f = \phi c_f/(1 - \phi)c_s$ (from Eqn 5, we see that $1/\zeta$ is a non-dimensional velocity). The non-dimensional versions of equations 1 and 2 are now (5) and (6):

$$\zeta \frac{\partial T'_f}{\partial t'} + \frac{\partial T'_f}{\partial x'} = -(T'_f - T'_s) + \left[ \frac{\lambda_f}{c_f} \frac{K_f}{v_{channel}^2} \right] \frac{\partial^2 T'_f}{\partial x'^2} \equiv -(T'_f - T'_s) + D_f \frac{\partial^2 T'_f}{\partial x'^2} \tag{5}$$

$$\frac{\partial T'_s}{\partial t'} = (T'_f - T'_s) + \left[ \frac{K_f}{K_s} \frac{\lambda_s}{c_s} \frac{K_f}{v_{channel}^2} \right] \frac{\partial^2 T'_s}{\partial x'^2} \equiv (T'_f - T'_s) + D_s \frac{\partial^2 T'_s}{\partial x'^2} \tag{6}$$

Analytic solutions for this set of equations have been derived for a number of limiting cases, particularly for $D_f = D_s = 0$ (Spiga and Spiga, 1981; Kuznetsov, 1994, 1995a, b, 1996), and were used to test the numerical calculations in this study.

The terms in square brackets in Eqns 5 and 6 represent dimensionless coefficients that govern the diffusion terms, $D_f$ and $D_s$. Using the definition of $\zeta$, we can further simplify the coefficient of the diffusion term $D_s$ and express it in terms of $D_f$ as

$$D_s = D_f \left[ \frac{1-\phi}{\phi} \frac{\lambda_s}{\lambda_f} \right] \tag{7}$$

It is clear that for a given temperature difference, $(T_f' - T_s')$, the behavior of Eqns 5 and 6 is governed by $D_f$ and $\zeta$ ($1/\zeta$ is a dimensionless in-channel velocity), thermal conductivities $\lambda_s$ and $\lambda_f$, and the channel volume fraction, $\phi$. We can define an effective Péclet number for the problem as the product of the velocity $v_{channel}$, the characteristic length scale $v_{channel}/K_f = v_{channel}(\phi c_f)/K$, divided by the thermal diffusivity of the channel material $\lambda_f/c_f$,

$$Pe_1 = \frac{v_{channel}^2 c_f^2 \phi}{\lambda_f K} = 1/D_f \tag{8}$$

where heat exchange due to thermal disequilibrium across channel walls (disequilibrium heating) will be important when $Pe_1 >> 1$. Since $K$ is a function of user-specified material and channel geometry parameters, we may further write $Pe_1$ as

$$Pe_1 = v_{channel}^2 \frac{c_f^2}{\lambda_f} \left[ \frac{1}{Nu\lambda_f} + \frac{1}{\beta\lambda_s} \right] \frac{d^2\phi}{A(1-\phi)} \tag{9}$$

Alternatively, one may also define an effective Péclet number as the product of $v_{channel}^2$, a characteristic timescale $1/K_s = c_s(1-\phi)/K$, divided by the thermal diffusivity of material outside channels $\lambda_s/c_s$,

$$Pe_2 = v_{channel}^2 \frac{c_s^2}{\lambda_s} \left[ \frac{1}{Nu\lambda_f} + \frac{1}{\beta\lambda_s} \right] \frac{d^2\phi}{A} \tag{10}$$

These definitions are limited to consideration of channelized flow and more complex dependence between $Pe$ and $\phi$ is suggested in the case of tubes or pores (e.g., Chevalier and Schmeling, 2022). Using values of $Nu = 12$, $\beta = 6$, $A = 2$, with $\phi$ and material properties in Table 1, for a given $v_{channel}$, $d$, $Pe_1$ is $\approx 2$ to $3$ times larger than $Pe_2$. Hereafter, I use the definition of $Pe_2$ when referring to Péclet number $Pe$.

Note that the definitions in Eqns 9 and 10 referred to as effective Péclet numbers, to distinguish them from the Péclet number one can define for axial transport in the channels, $Pe_{channel} = v_{channel} d c_f/\lambda_f$. By using the characteristic times $1/K_f$ or $1/K_s$, these definitions explicitly consider timescales associated with thermal exchange perpendicular to the transport direction, which depends on material parameters both inside and outside channels. In this manner, the definitions in Eqns 8 to 10 take into account the key role played by heat exchange governed by $K$ that works to reduce the thermal contrast across the channel walls.

## 2.3 Numerical method

Equations 5 and 6 are solved numerically in explicit time using a leap-frog method. The 1D domain $x' = [0, L]$ with $L = 10^2$ to $10^3$ is discretized typically with $N = 1000\text{-}5000$ elements and the maximum time for each run is $t'_{max} > L\zeta$ (for thermal perturbations considered below that have a characteristic time $\tau$, I use $t'_{max} > L\zeta + 2K_s\tau$). For a given element size $dx' = L/N$, $2^{nd}$ order finite differences are used for all spatial derivatives. The timestep $dt'$ is chosen (empirically) to be small enough to avoid numerical dispersion (for Pe > 1, it is sufficient to choose $dt' = dx' * 10^{-4}$). The code is written in Matlab (R2021b) and results below are confirmed using different grid resolutions. Step-function perturbation models below without axial diffusion ($D_f = D_S = 0$) were benchmarked (using leap-frog with upwind differencing for the spatial derivatives) against analytic solutions in Schumann (1929).

## 3 Results

In the models below, I consider a 1D domain ($x \geq 0$) that is initially in equilibrium $T'_s = T'_f = 0$, material outside the channels is stationary, and only the in-channel material is moving, at speed $v_{channel}$. At $x = 0$ and $t = 0^+$, the in-channel material is subjected to a thermal perturbation in $T_f$. The non-dimensional equations (5), (6), and (7) are used to determine the transient thermal equilibration between material inside and outside channels, but we limit consideration to models where advective transport dominates and $Pe > 10$. I consider transient thermal evolution with three scenarios of influx of hotter-than-ambient melt/fluid, in order of increasing complexity (Figure 1): (I) a step-function increase in temperature of the in-channel material; (II) a sinusodal temperature perturbation; and (III) a finite-duration, constant amplitude thermal pulse. The perturbation in $T_f$ disturbs the initial steady state condition in the domain starting at $t = 0^+$ as material with a perturbed temperature enters channels at $x = 0$ with a positive perturbation amplitude $\Delta T$ (Figure 1). The perturbation "front", the farthest extent of channel material that entered at $x = 0$ with perturbed temperature, is at $x_{\text{pert}} = v_{channel}t$ (or $x'_{\text{pert}} = t'/\zeta$; where the dimensionless velocity inside channels is $1/\zeta$; see 2.2). (Note that the location of maximum disequilibrium heat exchange $(T'_f - T'_s)_{\max}$) moves at a different speed, $V_{diseqm}$, discussed below.) In each case, the physical, user-specified quantities that specify the model are: the channel spacing, $d$, channel volume fraction $\phi$, the in-/out- channel thermal conductivities $\lambda_f$ and $\lambda_s$, heat capacitances, $c_f$, and $c_s$, and the speed of the in-channel material, $v_{channel}$ (see values in Table 1). Since the material properties $c_s$ and $c_f$ are held constant, there is a unique mapping between $\zeta$ and $\phi$, the channel volume fraction (for $\phi = 0.01$ to $0.20$, $\zeta = 0.0096$ to $0.2376$; Table 1).

## 3.1 Response to a step-function

The domain is initially at steady-state in equilibrium at temperature $T_s = T_f = T_0$ ($T'_s = T'_f = 0$) and at $t = 0$ the temperature of the in-channel material entering at the inflow $x = 0$ is perturbed so that, $T_f(x = 0, t > 0) = T_0 + \Delta T$ (or $T'_f = 1$). As material with higher temperature enters the channels, a transient thermal response occurs as the material inside and outside channels begin to equilibrate (material in channels cool, surroundings heat). A transient disequilibrium zone is observed to travel into

the domain (Figure 3a and b). Ahead of this disequilibrium zone, the channels are in equilibrium with the surroundings at the initial ambient temperature, $T_s' = T_f' = 0$. Behind this zone, the channels are in equilibrium with the surroundings at the inlet temperature, $T_s' = T_f' = 1$. The models compared in Figure 3a and b (same $\phi$ but different channel spacing $d$) show that the transient disequilibrium zone is observed to travel into the domain at the same $V_{diseqm}$, controlled by $\phi$, but the observed broadening of initially steep thermal profiles during transport differs, suggesting it is controlled primarily by the channel spacing $d$, and therefore the effective heat transfer coefficient $K$. Although diffusion plays a role in the broadening of the thermal profiles downstream, it is important to note that, unlike a simple advection-diffusion equation (where at large Péclet number we might not expect as much broadening of an initially sharp pulse), both terms on the right hand sides of Eqns 5 and 6 will drive broadening in this model. Even in the absence of diffusion therefore, thermal contrast between the material inside and outside channels causes shallowing of steep thermal gradients during transport (e.g., Figure 3c and d; see also Roy (2020)). At large $Pe$ this disequilibrium heat exchange dominates over axial conduction.

Following an initial lag time (when the maximum thermal contrast is at $x = 0$), the disequilibrium zone (marked by the peak in the $T_f' - T_s'$ function, $(T_f' - T_s')_{\max}$ in Figure 4b) migrates inward migration at a steady speed $V_{diseqm}$, a fixed fraction of $v_{channel}$ that depends on $\phi$, the channel volume fraction (Figure 4a and b). The ratio $V_{diseqm}/v_{channel}$ varies linearly with $\phi$ (Figure 4c). In the near-equilibrium limit, $(T_f' - T_s' \approx 0)$, Kuznetsov (1994) shows that the shape of the temperature difference function (Figure 4b) approaches a Gaussian with width that depends on $\sqrt{t'}$ and the zone of disequilibrium migrates at speed $v_{channel} c_f / (\phi c_f + (1 - \phi)c_s)$. Our models show that, when there is significant disequilibrium, the zone of disequilibrium migrates with a rate given by Eqn 11 (Figure 4c), independent of $K$, the heat transfer coefficient. Empirically, a key result is that the location(s) of maximum disequilibrium $(T_f' - T_s')_{\max}$ and therefore the greatest heat exchange progresses inward into the domain at a rate given by material properties, the channel volume fraction, $\phi$, and the in-channel velocity, $v_{channel}$,

$$V_{diseqm} \approx v_{channel} \left( \frac{c_f}{c_s} \right) \left( \frac{c_f \phi}{c_f \phi + (1 - \phi)c_s} \right) \tag{11}$$

Although $V_{diseqm}$ is independent of $K$, it is important to note that the degree of disequilibrium is not. Figure 3 illustrates the dependence on $K$ for the specific case where the in-channel velocity $v_{channel}$=1 m/yr, and $d = 500$ to $1000$ m, which correspond to $K = 4.7 \times 10^{-5}$ and $1.2 \times 10^{-5}$ W m$^{-3}$ K$^{-1}$, respectively. A second key result that emerges is that the degree of disequilibrium decreases exponentially as the zone of disequilibrium either migrates inward. This spatial decay is observed in Figure 3b, however it is quantified below, considering periodic thermal perturbations.

## 3.2   Response to a sinusoidal thermal perturbation

Here we consider a second scenario where the fluid entering the domain is hotter than the ambient initial temperature, but the thermal contrast varies sinusoidally. Although this is an idealized condition, it may be interpreted to represent periodic pulses of high temperature material entering into fluid- or melt-rich channels. Since any continuous time-varying thermal history at the inflow may be represented as a sum of sinusoids, this scenario also helps build intuition regarding the inherent length and timescales of equilibration. Sinusoidal thermal pulses introduce a new timescale into the problem, the period $\tau$, and the relevant

timescale to compare to is $1/K_s$ is the longest response timescale in the domain (of order $10^3$ yrs, see 2.1), associated with the thermal response of the material outside channels.

    Periodic thermal perturbations that might represent melt infiltration pulses lasting $10^4$ to $10^6$ yrs are characterized by a region of spatially varying temperatures: a thermal re-working zone (TRZ) (Figure 5). A key result is that thermal pulses with periods that are long compared to $1/K_s$ penetrate farther into the domain than shorter period oscillations (Figure 5). The

non-dimensional period, $\tau K_s = \tau K/(1-\phi)c_s$, controls the length scale, $\delta$, over which thermal oscillations penetrate into the domain. The wavelength of these temperature oscillations is set by the period $\tau$, $\lambda = v_{channel}\tau$. The penetration distance of the oscillations, $\delta$, is the maximum width of the TRZ. At short times, when $t \ll \delta/V_{diseqm}$, there is one zone of disequilibrium (the TRZ), bounded by the inward-traveling edge of the region of spatially-oscillating temperatures. The TRZ is initially narrow, but widens at a rate $V_{diseqm}$, eventually reaching a maximum width, $\delta$ (akin to a thermal "skin depth"), at time $\delta/V_{diseqm}$.

When $t \gg \delta/V_{diseqm}t$, there are two zones of disequilibrium: a stationary one bounded by the inlet with width $\delta$ (the TRZ, e.g., the wide region of gray at the left edge of the domain in Figure 5a and b), and a migrating zone that continues inward at $V_{diseqm}$ (inset in Figure 5b). The amplitude of the temperature oscillations decay with distance, but at each location in the TRZ the amplitude is constant, once oscillations are established (Figure 5). This effect is identical to the exponentially-decaying degree of chemical disequilibrium obtained using a very similar set of equations as 2 and 1 in the analytic solutions of Kenyon

(1990).

    As we might expect, the maximum width of the TRZ, $\delta$, is set controlled both by the non-dimensional oscillation period, $\tau/t_s$ and by the heat transfer coefficient, $K$. In the absence of the axial diffusion terms, previous work (Spiga and Spiga (1981); Kenyon (1990)) has shown that,

$$\delta = \left(\frac{c_f\phi v_{channel}}{K}\right)\left(\frac{(\tau/t_s)^2}{4\pi^2}\right) \tag{12}$$

noting that $K \sim d^{-2}$ (2.1), and $t_s = 1/K_s = c_s(1-\phi)/K$, the expression above suggests that, for fixed $v_{channel}$, $\delta$ is propor-

tional to $(\tau/d)^2$ in the absence of axial conduction. With axial conduction, in the limit of large $d$ (large heat transfer coefficient and Péclet number), this scaling is confirmed by the numerical results for sinusoidal periodic perturbations (square symbols in Figure 7). A consequence of the inward-decaying degree of disequilibrium between material inside and outside of channels is that disequilibrium heat exchange occurs only within a certain distance, $\delta$, of the inlet ($x = 0$), within the TRZ (Figure 5).

### 3.3   Finite pulse, duration $\tau_p$, amplitude $\Delta T$

Here we consider the fate of a finite-duration thermal pulse, representing episodic infiltration of melts that are hotter than the ambient CLM. This scenario introduces a timescale into the problem, the pulse duration $\tau_p$, implemented here as a tanh-function $(1/2)(\tanh(\frac{(t-\tau_p/2))}{w}) - \tanh(\frac{(t-3\tau_p/2)}{w}))$ with a characteristic growth/decay timescale $w$ that depends upon $\tau_p$, e.g., $w = 0.1*\tau_p$ in these models. The results for a finite duration pulse, exemplified in Figure 6, are consistent with the characteristic transient behavior already apparent in the sine and step function perturbations above. The thermal perturbation

$T_f'$ in the channels at $x = 0$ km (the inlet) is distorted as it proceeds into the domain, broadening in width and decaying in

amplitude (Figure 6a). Similarly, the zone of disequilibrium heat exchange (marked by $(T'_f - T'_s)_{\max}$), migrates into the domain at a rate given by Eqn 11 and decays during transport (Figure 6b).

As with the sinusoidal perturbation, these effects lead to a TRZ that widens over time to a maximum width, $\delta$, that scales with the duration of the thermal pulse $\tau$, channel spacing $d$, and volume fraction $\phi$. Figure 7 illustrates how, for fixed $v_{channel} = 1$ m/yr, independent parameters $d$, $\phi$, and $\tau$ lead to variable TRZ widths, $\delta$. For a given channel spacing $d$, which strongly controls the heat transfer coefficient, $K \sim d^{-2}$ (large $d$, small $K$), I consider a range of plausible channel volume fractions $\phi = 0.01$ to $0.2$ (Table 1). Whereas $\delta$ is proportional to $\tau^2$ for large $d$ in the pure sinusoid case (square symbols in Figure 7), the finite-duration pulse results point to an $n$ value likely less than 2, closer to 1 (Figure 7). This is likely due to the fact that sinusoidal perturbations represent a higher, and continuous energy input into the system compared to a finite-duration pulse, suggesting that in the case of multiple (episodic) pulses of melt-infiltration, the scaling exponent is likely to be between 1 and 2. To obtain a TRZ width between 1 to 10 km, we require channel spacings around $d = 100$ (with $\phi = 0.1$ to $0.2$), $d = 500$ m (with $\phi < 0.16$), or $d = 1000$ m (with $\phi < 0.02$) and thermal pulse durations of $10^1$ to $10^2$ Kyrs (Figure 7).

## 4 Discussion

The model scenarios considered above are idealized and therefore limited in their representation of the complexities of deformation and fluid-rock interactions within the CLM. In particular, the effective thermal properties and the geometry of the fluid-/melt-rich channels are abstracted into a single number, the heat transfer coefficient, $K$, strongly controlled by the channel spacing, $d$. Sinuosity and other aspects of the geometry of channelization are abstracted and the details of processes at and below the scale of an average channel spacing, $d$, are ignored. Instead, the focus here is on the effective behavior at mesoscopic spatial scales $\gg d$. Even at these scales, this formulation ignores spatial variations in transport, including variability in the channel volume fraction $\phi$, in-channel velocity $v_{channel}$, and effective heat transfer coefficient $K$. Time-dependent variability in transport is also ignored, e.g., feedbacks due to possible phase changes during disequilibrium heating/cooling which would affect the geometry of the channels (Keller and Suckale, 2019). Finally, this 1D model ignores the 3D nature of relative motion between material inside and outside channel walls even on the mesoscale ($\gg d$).

Given these limitations, the models above are a way to frame first-order questions and develop arguments related to the consequences of disequilibrium heating, particularly when the behavior is dominated by downstream effects in the direction of transport. Taking the model domain to be analogous to the lowermost lithosphere, where melt or fluid transport may be channelized (Figure 8), $x < 0$ corresponds to a melt-rich sub-lithospheric region (e.g. a decompaction layer, Holtzman and Kendall, 2010), whereas the domain $x > 0$ represents an initially sub-solidus lowermost CLM, and $x = 0$ is the initial LAB (Figure 8). Melt-infiltration into the lithosphere may be episodic, controlled by timescales associated with transport from the melt-generation zone to the LAB (e.g., Scott and Stevenson, 1984; Wiggins and Spiegelman, 1995), processes of fracturing and crystallization in a dike boundary layer (e.g., Havlin et al., 2013) and melt supply from a deeper region of melt production (e.g., Lamb et al., 2017).

Three key results emerge from the models above: (1) disequilibrium heating, estimated using the heat transfer coefficient, may be a significant portion of the heat budget at the LAB and the lowermost CLM, (2) there is a material-dependent velocity associated with transient disequilibrium heating, and (3) there is a region of spatio-temporally varying disequilibrium heat exchange, a thermal reworking zone (TRZ). Below I discuss each of these within the context of episodic melt-infiltration into the CLM in an intra-plate setting, specifically the Basin and Range province of the western US where deformation and 3D melt transport may be simplified by neglecting plate-boundary effects. In this case, dominantly vertical heat transport within a slowly deforming lithosphere is a reasonable first-order assumption.

**i. Disequilibrium heating and the heat budget at the LAB.** The relative importance of disequilibrium heat exchange at the LAB may be established by considering the effective heat transfer coefficient, $K$, and the parameter which most strongly controls it, namely the average spacing of channels, $d$. For the material parameters in Table 1, and channel volume fraction $\phi \approx 0.1$, channel spacing of $d = 500$ to $1000$ m, $K$ is of order $10^{-5}$ W m$^{-3}$K$^{-1}$ (2.1), corresponding to Péclet numbers of order $10^1$. Physically, $K$ corresponds to across-channel heat transfer per unit time, per unit volume, per unit difference in temperature (Schumann, 1929). Therefore, if we assume that the average thermal contrast in the TRZ is roughly 2 to 5 % of $\Delta T$ (e.g., $(T'_f - T'_s)_{\max}$ in Figure 6), for $\Delta T = 100$ K excess temperature of the infiltrating melt, disequilibrium heat exchange might contribute around $10^{-4}$ to $10^{-5}$ W m$^{-3}$ to the heat budget at the LAB. This is a conservative estimate, given that the temperature difference between magma and the surrounding material may be larger (e.g., in Lherz the inferred contrast is $>200$ K (Soustelle et al., 2009); and up to 1000 K in crust; (Lesher and Spera, 2015)). Similarly, plume excess temperatures are estimated to be as large as 250 K (Wang et al., 2015).

To put this in perspective, we now compare this estimated heat budget to the heat budget due to deposition of latent heat during crystallization of melt transported in channels in the lithosphere (e.g., Havlin et al., 2013; ReesJones et al., 2018). The contribution from freezing of melt may be estimated using scaling arguments made in Havlin et al. (2013). Assuming that melt and rock are in equilibrium, Havlin et al. (2013) estimate that the heat released by a crystallization front would contribute around $\rho H S_{dike}$, where $\rho$ is the melt density, $H$ is the latent heat of crystallization, $S_{dike}$ is a volumetric flow rate out of a decompacting melt-rich LAB boundary layer due to diking. For a representative dike porosity of $\phi = 0.1$ within the dike, Havlin et al. (2013) estimate $S_{dike} \approx 0.2$ mm$^3$/s. Taking $\rho = 3000$ kg m$^3$, and $H = 3 \times 10^5$ J/kg, the heat source due to the moving crystallization front would be around $10^2$ W per dike. If we assume that this heating takes place within a volume that is roughly the dike height $\times$ dike spacing $\times$ dike length, we can determine the volumetric power generated due to crystallization. For example, assuming dike heights of about 1 km and dike spacing large enough for non-interacting dikes (as estimated by Havlin et al. (2013), a porosity of 0.1 would require a dike spacing of $\sim 1$ km), the heat source due to a crystallizing dike boundary layer would be $< 10^{-4}$ W m$^{-3}$ (per unit length along strike). These arguments corroborate the idea that disequilibrium heat exchange during melt-rock interaction could be an important portion of the heat budget at the LAB as compared to other expected processes, such as heating due to crystallization of melt in channels (see also ReesJones et al., 2018).

**ii. Progression of a disequilibrium heating zone/front at a rate $V_{diseqm}$.** The disequilibrium heating front is associated with a migration rate $V_{diseqm}$ that is less than the in-channel material velocity, $v_{channel}$ (Eqn 11). Therefore, $V_{diseqm}$ limits

the rate at which the lowermost CLM may be modified by thermal disequilibrium during migration of a disequilibrium front (e.g., Figures 5b, 6). Although in this work I considered a fixed $v_{channel} = 1$ m/yr, note that $V_{diseqm}$ (Eqn 11) is independent of temperature contrast between the CLM and infiltrating melt and, for fixed material properties and channel volume fraction $\phi$, depends linearly on $v_{channel}$. However, Eqn 11 also illustrates a tradeoff between $v_{channel}$ and $\phi$. Assuming a channel volume fraction of $\phi = 0.1$ at the LAB, and material properties in Table 1, $V_{diseqm}$ would be around 10 % of the in-channel velocity (Figure 4c). For in-channel velocity in the range of 0.01 to 1 m/yr, the disequilibrium heating front at the LAB would migrate upward at a rate of $\approx 1$ to $10^2$ km/Myr (Figure 8) This is comparable to rates of CLM thinning predicted by heating due to the upward motion of a dike boundary layer (1 to 6 km/Myr in Havlin et al. (2013)). Interestingly, an upward-moving disequilibrium heating zone with $V_{diseqm} \approx 1$ to $10^2$ km/Myr also brackets the 10-20 km/Myr rate of upward migration of the LAB inferred from the pressure and temperature of last equilibration of Cenozoic basalts in the Big Pine volcanic Field in the western US (Plank and Forsyth, 2016). An implication of the models here, therefore, is that disequilibrium heating may produce lithosphere modification at geologically-relevant spatial and temporal scales provided that the material velocity in channels at the LAB is on the order of $10^{-1}$ to 1 m/yr (e.g., Figure 8; Rutherford, 2008); higher transport rates would require lower $\phi$ to drive a similar rate of CLM modification.

**iii. Thermal reworking zone (TRZ).** A key result that may be relevant to the evolution of the LAB is that episodic infiltration of melts that are hotter than the surrounding CLM would lead to a finite region of disequilibrium heating within a thermal reworking zone or TRZ. The TRZ would undergo a phase of transient widening (at a rate given by $V_{diseqm}$), reaching a maximum width $\delta$ that is proportional to $[\phi v_{chan}(\tau/d)^n]$ ($n \approx 1$ for individual perturbations, but $< 2$ for multiple; Figure 7; also 5d). Here $d$ is a characteristic scale of channelization and $\tau$ is a timescale associated with the episodicity of melt-infiltration. This scaling gives us a way to conceptualize the modification of the lowermost CLM as a zone that may encompass a variable thickness TRZ, depending on variability in transport velocity and in the timescale of melt-infiltration (Figure 8b). Regions where the timescale of episodic melt-infiltration is longer are predicted to have a thicker zone of modification at the LAB (Figure 7). For example, for $\phi \approx 0.12$, a channel spacing of $d = 500$ m, disequilibrium heating by melt pulses that last around 10 Kyrs implies a maximum thickness of roughly 10 km for the zone of modification (Figure 7; also 5d). In this scenario, the TRZ grows to this maximum width over a timescale governed by $\delta/V_{diseqm}$; for $V_{diseqm} = 10$ km/Myr, which corresponds to melt velocity of roughly 0.1 m/yr (see (ii) above), the 10 km wide TRZ would be established within about 1 Myr (Figure 8), comparable to rates of CLM modification inferred from observations in Plank and Forsyth (2016).

These scaling arguments lead to the idea that perhaps the TRZ represents a zone of thermal modification at the base of the CLM that may also correspond to (or encompass) a zone of rheologic weakening and/or in-situ melting if the infiltrating fluids are hotter than the ambient material. The dynamic evolution of the LAB during episodic pulses of melt-infiltration is beyond the scope of the simple models above (which assume a stationary, undeforming matrix). However, assuming mantle material obeys a temperature and pressure-dependent viscosity scaling relation such as in Hirth and Kohlstedt (2003) at an LAB depth of about 75 km, where the ambient mantle is cooler than the dry solidus (e.g., 1100ºC + 3.5ºC/km; Plank and Forsyth (2016) and assuming a wet dislocation creep mechanism), we would expect a viscosity reduction by factor $\approx 1/1.4$ to $1/12$ during heating (e.g., $\approx 1/2.3$ for a temperature increase of 20 K ($0.2\Delta T$ for a 100 K perturbation amplitude)). This effect is weaker,

but still important for a deeper LAB; e.g, at 125 km depth, the viscosity reduction would be a factor $\approx 1/1.2$ for a temperature increase of 20 K.

Geochemical evidence from Cenozoic basalts from the western US, particularly space-time variations in volcanic rock Ta/Th and Nd isotopic compositions suggest that the timescale of modification and removal of the lowermost CLM is on the order of $10^1$ Myrs (Farmer et al., 2020). These authors argue that the observed transition from low to intermediate to high Ta/Th ratios indicates a change from: arc/subduction-related magmatism, to magmatism associated with in-situ melting of a metasomatized CLM (the "ignimbrite flare-up"), to magmatism due to decompression and upwelling after removal of the lowermost CLM. At
a minimum, the observed timescale of the transition in Ta/Th ratios in volcanic rocks ($10^1$ Myrs) in the western US should be comparable to the timescales of degradation of the CLM. If correct, these interpretations and observations are promising and provide an important avenue for exploring the role of thermal and chemical disequilibrium during melt-rock interaction and destabilization of the CLM in an intra-plate setting.

## 5    Conclusions

In summary, I have presented arguments supporting the role of disequilibrium heating in the modification of the base of the continental lithospheric mantle (CLM) during melt-infiltration into and across the lithosphere-asthenosphere boundary (LAB). Infiltration of pulses of hotter-than-ambient material into the LAB should establish a thermal reworking zone (TRZ) associated with disequilbrium heat exchange. The spatial and temporal scales associated with the establishment of the TRZ are comparable to those for CLM modification inferred from geochemical and petrologic observations intra-plate settings, e.g., the western US.
Disequilibrium heating may contribute around $10^{-5}$W/m$^3$ to the heat-budget at the LAB and, for transport velocity of 0.1 to 1 m/yr in channels that are roughly $10^2$ m apart, a 10 km wide TRZ may be established within 1 Myr. Disequilibrium heating during melt-infiltration may therefore be an important process for modifying the CLM. Further work is needed to explore its role in the rheologic weakening that must precede mobilization (and possibly removal) of the lowermost CLM.

*Code availability.* The codes (written in Matlab) used for the non-dimensional system presented are available at github.com/mousumiroy-
unm/diseqm22.

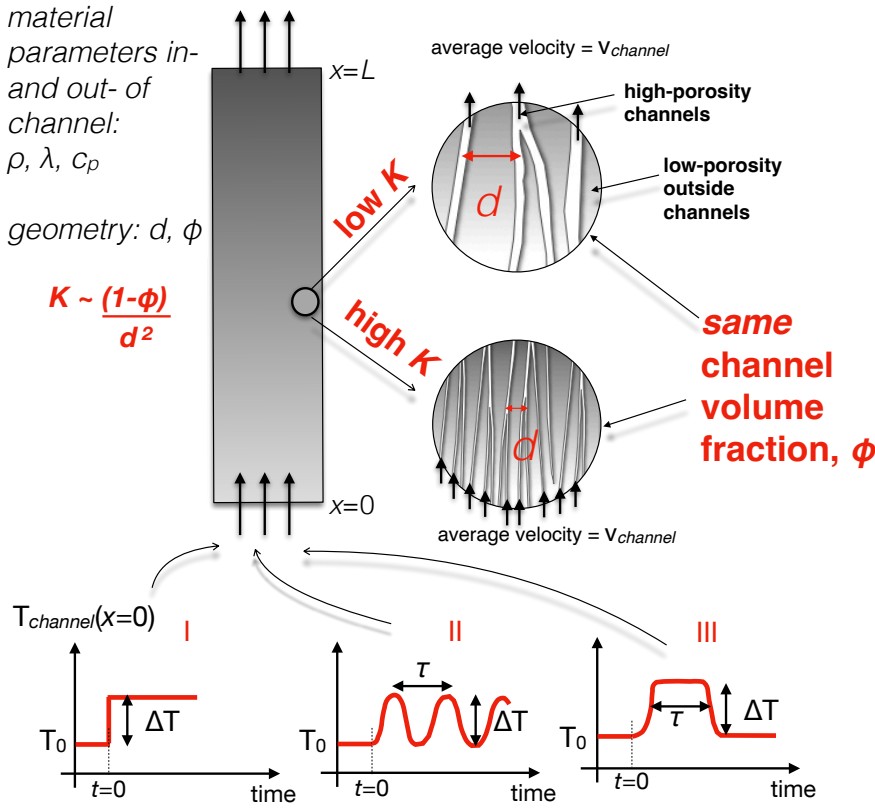

**Figure 1.** Cartoon of 1D model specified by: (1) material parameters within and outside of channels (heat capacities $c_{pf}$ and $c_{ps}$, thermal conductivities $\lambda_f$ and $\lambda_s$, and densities $\rho_f$ and $\rho_s$), (2) average in-channel velocity $v_{channel}$, and (3) geometric parameters such as channel volume fraction, $\phi$, and channel spacing $d$. The effective heat transfer coefficient $K$ is a function of heat capacitances ($c_{pf}/\rho_f$, $c_{ps}/\rho_s$) $\phi$, and $d$, scaling as $d^{-2}$ (see 2.1); large $K$ corresponds to large channel wall area per unit volume (e.g., small $d$) and vice versa. The input in-channel temperature vs. time functions considered in this study are shown in the three graphs below: (I) Step function; (II) Sinusoid; (III) Finite-duration pulse.

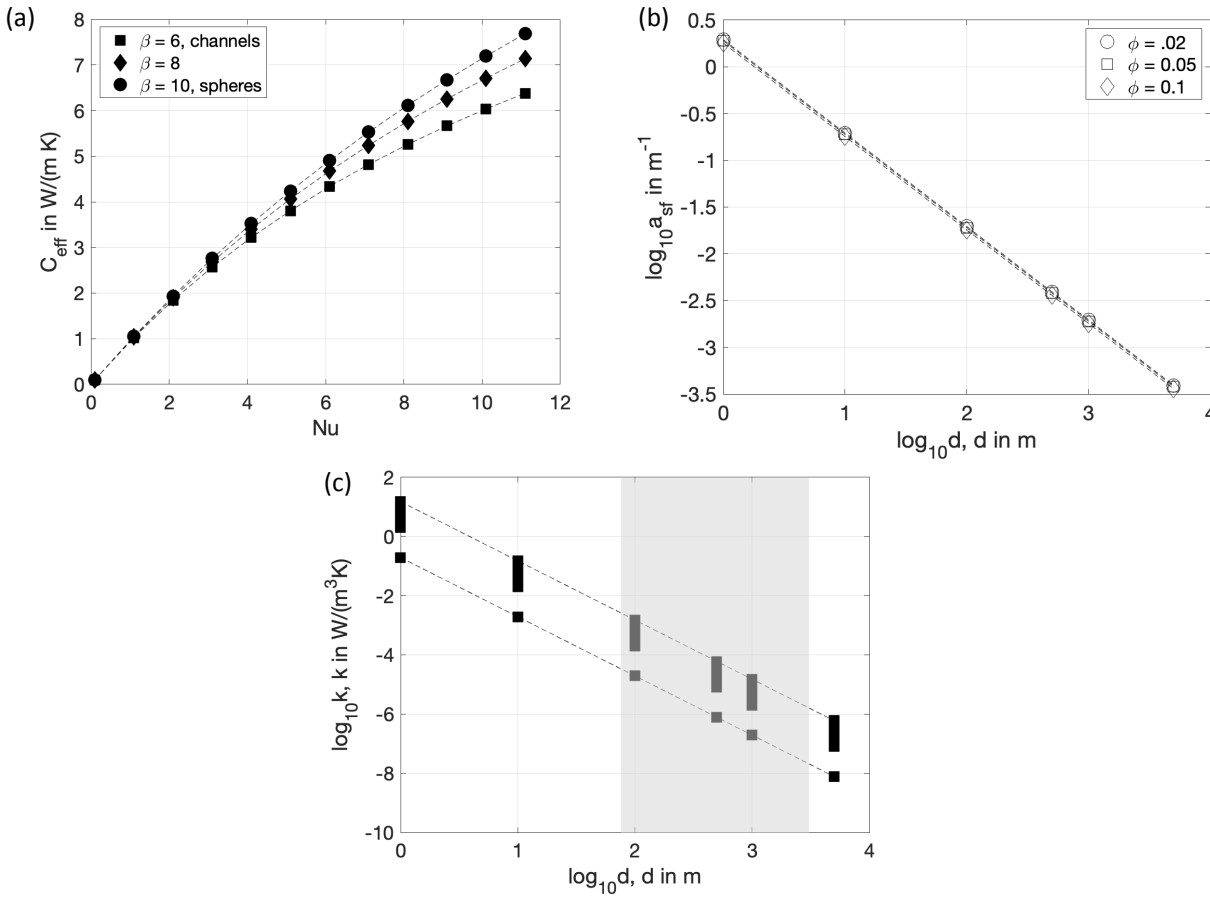

**Figure 2.** (a) Effective thermal conductivity, $C_{eff}$ in Eqn 3, as a function of Nusselt number, (b) geometric factor, $a_{sf}$, as a function of channelization scale, $d$, and (c) heat transfer coefficient, $K$, as a function of channelization scale $d$. For a fixed $d$, the dashed lines in (c) delineate the variation in $K$ for the range of $\beta$ values in (a) and $\phi$ values in (b), illustrating that $K$ is mainly controlled by $d$, rather than the other parameters. Gray shading indicates the range of channel spacings relevant to models with $Pe > 10$ presented in this study.

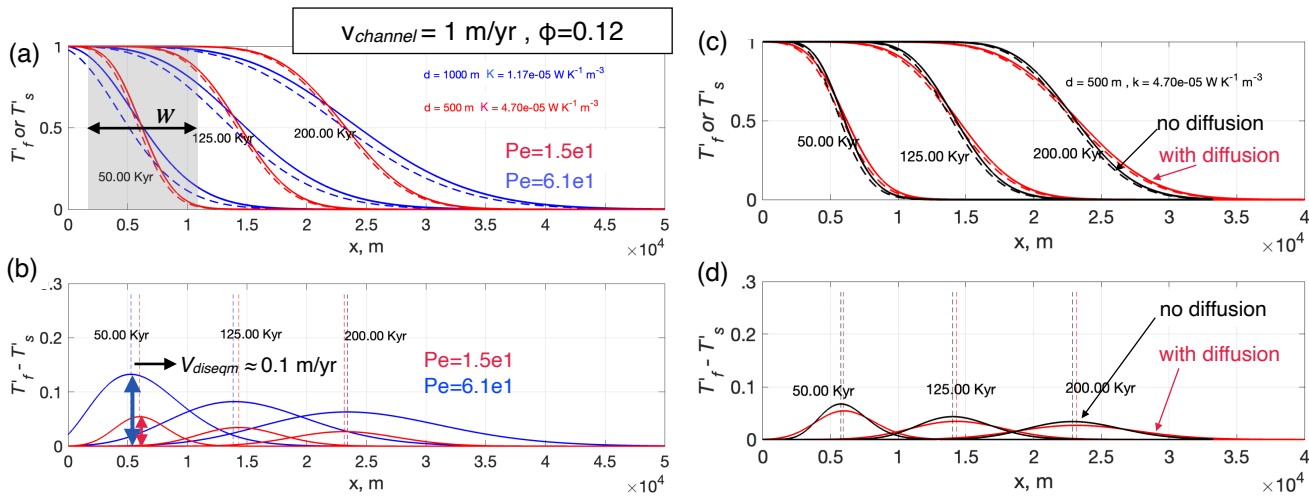

**Figure 3.** (a) Normalized temperature profiles, $T'_s$ (dashed lines) and $T'_f$ (solid lines) for models with in-channel velocity $v_{channel} = 1$ m/yr, at times $t = 50$, 125, and 200 Kyr after a step function perturbation in $T'_f$. The temperature profiles transition between the incoming channel material normalized temperature (=1, left) and the initial ambient temperature (=0, right). We compare temperature profiles for two cases with the same channel volume fraction $\phi = 0.12$ but different channel spacing, $d$, and therefore heat transfer coefficient $K$ and Péclet number, indicated. The transition region (e.g., highlighted in gray for the $d = 500$ m model at $t$=50 Kyr), has width, $w$, that is larger for smaller $K$ (large $d$) and increases over time. (b) Profile of the temperature difference $T'_f - T'_s$ between channels and surroundings at times indicated, for the same calculations in (a). The degree of disequilibrium (red and blue arrows) is greater for smaller $K$ (higher Pe) and decreases as a function of time. (c) and (d) compare the results for $d = 500$ m (red lines in (c) and (d) are identical to those in (a) and (b)), but using a modified model where the axial diffusion terms in Eqns 5 and 6 are neglected ($D_f = D_s = 0$).

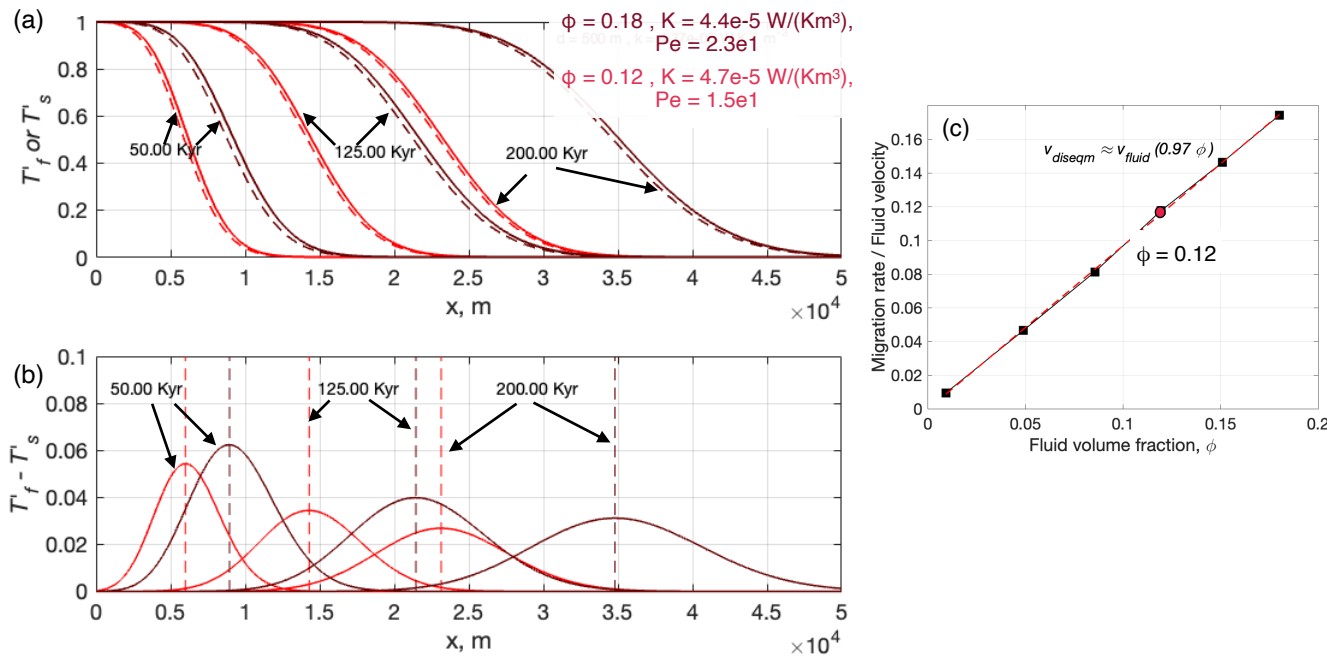

**Figure 4.** (a) Normalized temperature within channels (solid lines) and surroundings (dashed lines) at different times (indicated) following a step function perturbation in $T_f$. The models shown here have the same in-channel velocity $v_{channel} = 1$ m/yr and channel spacing $d = 500$, and are compared at times $t = 50$, $125$, and $200$ Kyr as in Figure 3. The two cases here differ, however, in the channel volume fraction, $\phi$ (indicated), illustrating that the migration rate of the zone of disequilibrium is a function of $\phi$. (b) Normalized temperature difference $T_f' - T_s'$ between channels and surroundings as a function of position, shown for the same times as in (a). (c) Normalized migration rate of the zone of disequilibrium as a function of channel volume fraction, $\phi$. Red dot is for $\phi = 0.12$, corresponding to bright red lines shown in (a) and (b), and in Figure 3.

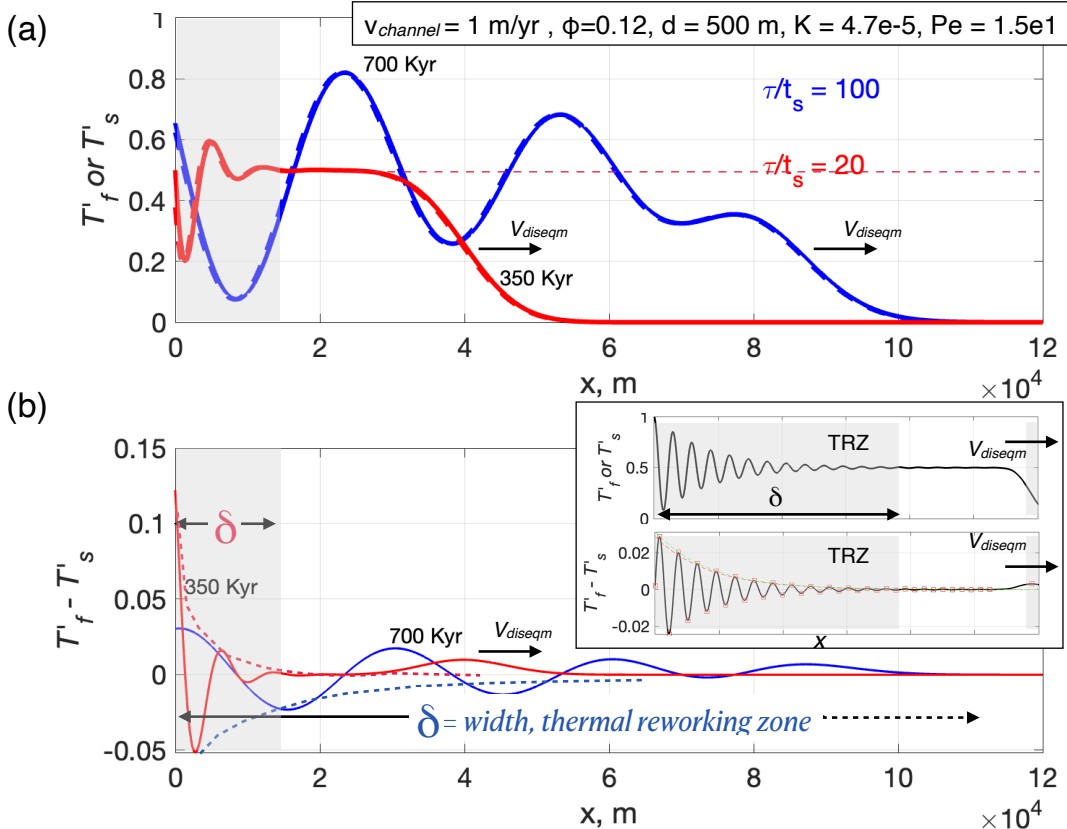

**Figure 5.** (a) Normalized temperature profiles in-channel $T_f'$ (solid lines) and out-of-channel $T_s'$ (dashes), at times indicated, for a calculation with in-channel velocity $v_{channel}$=1 m/yr, channel volume fraction $\phi$=0.12, channel spacing $d$=500 m and heat transfer coefficient $K$ and $Pe$ as indicated. For the chosen parameters, the response timescale is $1/K_s$=$t_s \approx 2.6$ Kyr. Results are shown for two different sinusoidal thermal variations in the incoming in-channel material with (normalized) oscillation periods: $\tau/t_s$=20 (red, shown at $t = 350$ Kyr) and $\tau/t_s$=100 (blue, shown at $t = 700$ Kyr). The region of sinusoidal thermal profile (gray shading) has reached a steady-state width for the $\tau/t_s$=20 case but not for $\tau/t_s$=100. (b) Normalized temperature difference $T_f' - T_s'$ between channels and surroundings as a function of position, shown for the same times as in (a). The thermal reworking zone (TRZ) (grey shaded region for $\tau/t_s$=20) has spatial oscillations in $T_f' - T_s'$, with amplitudes that decrease over a length scale $\delta$, the width of the TRZ: $\delta$ is larger for longer period (blue) and shorter for shorter period (red). The degree of disequilibrium ($T_f' - T_s'$) is oscillatory in the TRZ, with decaying amplitude over width $\delta$. Inset: Cartoon illustrating a steady-state TRZ (once it has widened to a width $\delta$). During the widening phase, the TRZ width increases at a rate $V_{diseqm}$. Once it has grown to maximum width, $\delta$, beyond $x = \delta$ the domain is at equilibrium ($T_s'$=$T_f'$=0.5), except for a region that continues to migrate inward at $V_{diseqm}$ (arrows).

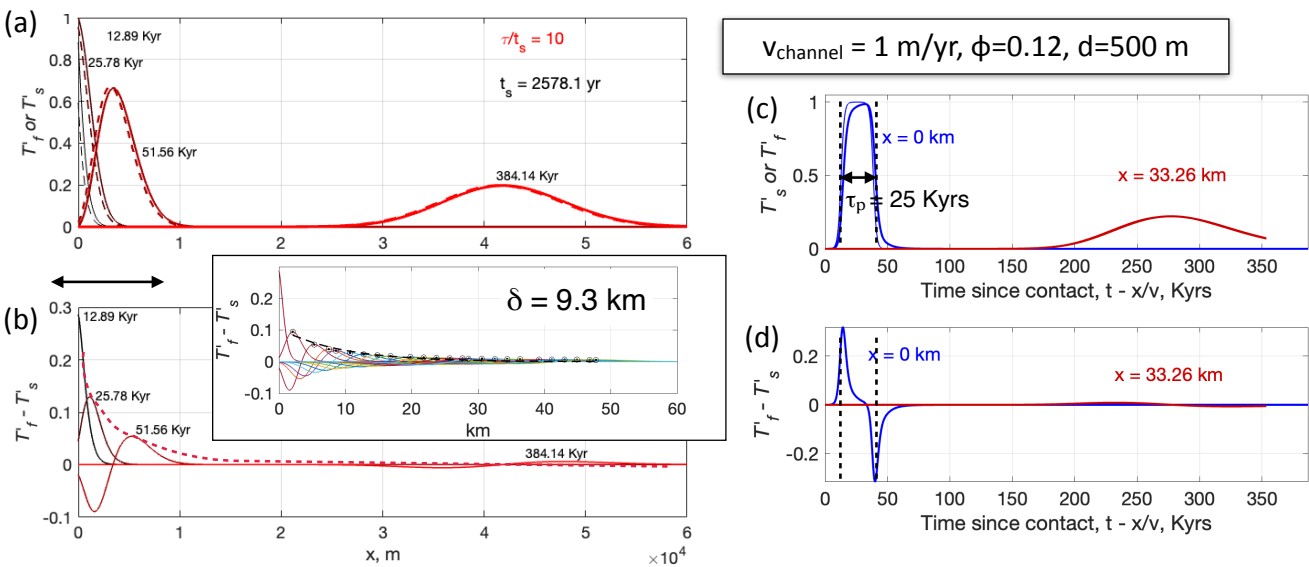

**Figure 6.** Two views of the thermal response to a finite-duration perturbation ($T_f'(x=0,t)$ is a $\tanh$-function) with duration $\tau_p = 25$ Kyrs, in a model with $v_{channel} = 1$ m/yr, channel volume fraction $\phi = 0.12$, and channel spacing $d = 500$ m. (a) Normalized temperature outside channels ($T_s'$, solid lines) and within channels ($T_f'$, dashed lines) at various times as indicated. (b) The degree of disequilibrium, $T_f' - T_s'$ as a function of position, the same times as in (a). Curved dashed line illustrates an exponentially-decaying envelope that is used to estimate $\delta$, the width of the TRZ. For clarity, intermediate temperature profiles used to estimate $\delta$ are shown in the inset. (c) Temperature-time history for a model with for a model with in-channel velocity Normalized temperature outside channels ($T_s'$, solid lines) and within channels ($T_f'$, dashed lines) at varying locations within the domain ($x$, as indicated) are plotted as a function of time since contact with material that entered channels at $x=0, t=0$. (d) The degree of disequilibrium, $T_f' - T_s'$ for the same $x$ locations as in (c), plotted as a function of time since contact with material that entered channels at $x=0, t=0$.

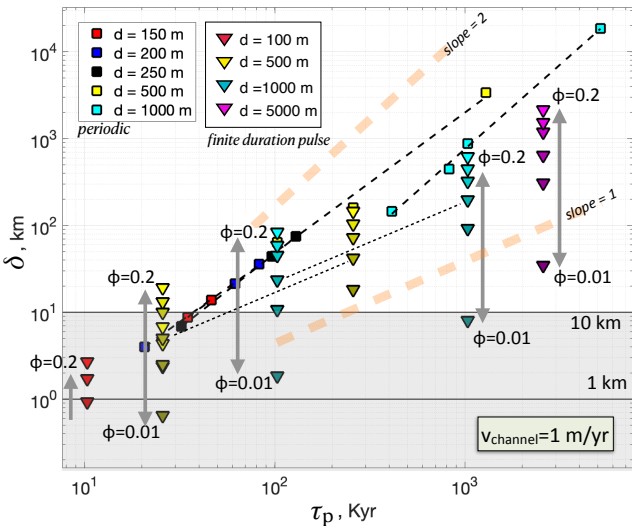

**Figure 7.** Width of the TRZ, $\delta$ for calculations with $v_{channel} = 1$ m/yr transport velocity and variable channel spacing, $d$. Numerically derived values of $\delta$ are obtained by fitting an exponential decay to the maximum degree of disequilibrium $\max(T'_f - T'_s)$, e.g, Figure 6b. Results compiled here are for periodic thermal perturbations with period $\tau_p$ (squares) and for finite-duration thermal perturbations with duration $\tau_p$ (triangles). For clarity, models with fixed $\phi = 0.12$ and varying $d$ are shown for the periodic perturbations. For finite duration pulses, results shown include variable $d$ and $\phi$, as indicated. For given $d$ and $\tau_p$, $\delta$ is a function of channel volume fraction $\phi = 0.01$ to $0.2$ (Table 1); the color saturation of the symbols corresponds to $\phi$: darkest=$\phi_{min} = 0.01$ and lightest=$\phi_{min} = 0.2$. A few representative thin black dashed lines are drawn to connect models that have the same channel geometry parameters $d$ and $\phi$, but differ only by the incoming thermal pulse duration, $\tau_p$. The slopes of the thin dashed lines therefore illustrate how $\delta$ scales with $\tau_p$. Light orange dashed lines show the slope expected for the analytic scaling in Eqn 12 (slope=2) and the approximate scaling observed here (slope=1) for finite-duration pulses. Thick horizontal lines indicate $\delta = 1$ and 10 km; gray shading represents $\delta \leq 10$ km, e.g., the lowermost 10 km of the CLM.

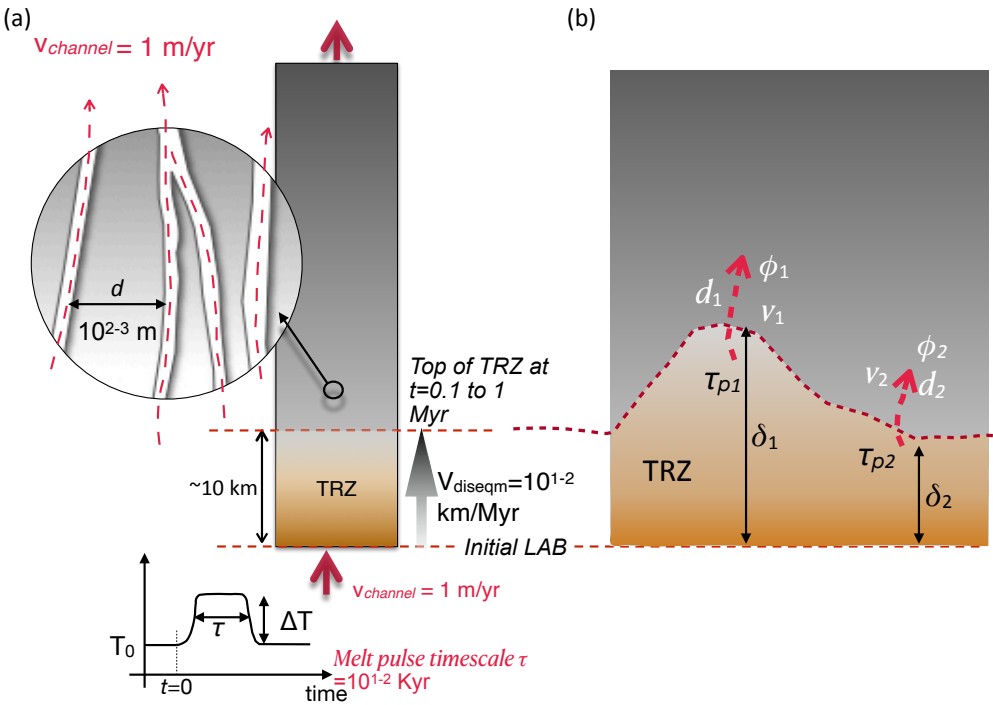

**Figure 8.** Implications for a thermal re-working zone (TRZ) that forms a modified layer at the lowermost CLM as a result of disequilibrium heating. (a) Cartoon illustrating a specific set of parameters that lead to TRZ growth to a steady-state width of $\delta \approx 10$ km, after 0.1 to 1 Myr: duration of melt-infiltration $\tau = 10^1$ to $10^2$ Kyrs, channel spacing $d$=500 to 1000 m, and in-channel material velocity $v_{channel}=$ 1 m/yr. The TRZ is characterized by upward-decreasing degree of disequilibrium (indicated by the color). Before reaching its final width, the TRZ transiently grows at a relatively fast rate, $V_{diseqm} >$10 km/Myr. (b) Cartoon illustrating an interpretation of spatially variable TRZ width, $\delta(x)$, as due to lateral juxtaposition of multiple regions of dominantly vertical (1D) transport, but with spatially variable $d$, $\phi$, $v_{channel}$, and $\tau_p$. In this case, $\delta_1 > \delta_2$, which could be due to $d_1 < d_2$, or $\phi_1 > \phi_2$ or $v_1 > v_2$, or $\tau_{p1} > \tau_{p2}$, or some combination of these.

**Table 1.** Material properties and constants used in calculations

| Name | Symbol | Value or range | Source/Comments |
|---|---|---|---|
| Melt, grain density | $\rho_{melt}, \rho_{grain}$ | 2800, 3300 kg/m$^3$ | Lesher and Spera (2015) |
| Melt specific heat capacity | $c_{pmelt}$ | 1400 J/(kg K) | Lesher and Spera (2015) |
| Grain specific heat capacity | $c_{pgrain}$ | 1250 J/(kg K) | Lesher and Spera (2015) |
| Melt heat capacity per volume | $c_{melt}$ | $3.920 \times 10^6$ J/(m$^3$ K) | $c_{pmelt} \times \rho_{melt}$ |
| Grain heat capacity per volume | $c_{grain}$ | $4.125 \times 10^6$ J/(m$^3$ K) | $c_{pgrain} \times \rho_{grain}$ |
| Melt thermal conductivity | $\lambda_{melt}$ | 1 W m$^{-1}$ K$^{-1}$ | Lesher and Spera (2015) |
| Grain thermal conductivity | $\lambda_{grain}$ | 2.5 W m$^{-1}$ K$^{-1}$ | Lesher and Spera (2015) |
| In-channel, out-of-channel grain-scale porosity | $\varphi_f, \varphi_s$ | 1, 0 | End-member case maximizing material property contrast (Appendix **??**) |
| In-channel heat capacity per volume | $c_f$ | $3.920 \times 10^6$ J/(m$^3$ K) | $c_f = \varphi_f c_{melt} + (1-\varphi_f)c_{grain} = c_{melt}$ |
| Out-of-channel heat capacity per volume | $c_s$ | $4.125 \times 10^6$ J/(m$^3$ K) | $c_s = \varphi_s c_{melt} + (1-\varphi_s)c_{grain} = c_{grain}$ |
| Heat transfer coefficient | $K$ | $10^{-5}$ to $10^{-7}$ W/m$^3$K | this work (section 2.1) |
| Channel volume fraction | $\phi$ | 0.01 to 0.2 | (e.g., Pec et al., 2017) |
| Channel average (linear) velocity relative to surroundings | $v_{channel}$ | 1 to 100 mm/yr | (e.g., Rutherford, 2008) |
| Weighted heat capacitance ratio | $\zeta$ | 0.0096 to 0.2376 | calculated, $\zeta = \phi c_f / (1-\phi)c_s$ |
| Fluid-solid Nusselt number | $Nu$ | 0.1 to 12.4 | for slow flows Handley and Heggs (1968) |
| Constant in Eqn 3 | $\beta$ | 6-10; use 6 here | Dixon and Cresswell (1979) |
| Constant in Eqn 4 | $A$ | may be 2-6; use 2 for channels here | Dullien (1979) |
| Separation of fluid-rich channels | $d$ | $10^2$ to $10^3$ m | LeRoux et al. (2008) |

*Author contributions.* This study was conceived, executed, and completed by Mousumi Roy

*Competing interests.* No competing interests are present.

*Acknowledgements.* This work benefited from detailed, thoughtful, and constructive reviews from H. Schmeling and an anonymous referee. I have greatly enjoyed conversations with L. Farmer, A. Clark, and R. Carlson regarding applications of the simple 1D model presented here to the western US and generally to melt-rock interactions in the real world. I am also grateful for discussions with C. Havlin and B. Holtzman during early stages of thinking about the 1D Schumann model. Funding sources for developing the model and ideas presented include: NSF EAR-0952325, EAR-2120812, an Aspen Center for Physics Fellowship, and a Women in STEM award from UNM-Advance. This study draws upon previously published observations and data/observations were not generated for this research.

**References**

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
