# Peer review of "Assessing the role of thermal disequilibrium in the evolution of the lithosphere-asthenosphere boundary: An idealized model of heat exchange during channelized melt-transport"

_Solid Earth, 2022_

## Referee Comment (RC1)

**Review of "Assessing the role of thermal disequilibrium in the evolution of the lithosphere-asthenosphere boundary: An idealized model of heatexchange during channelized melt-transport"**

Harro Schmeling

March 29, 2022

In this manuscript the thermal effect of melt infiltration into the base of the continental lithosphere is studied focusing on the thermal disequilibrium between melt and ambient rock. While thermal disequilibrium in porous flow is well studied in more technical literature, only a few papers quantitatively addressed this effect in the recent geoscience literature. Therefore this paper is timely and new. It is shown that indeed thermal disequilibrium may be important under certain circumstances near the lithosphere asthenosphere boundary explaining some observational data. Useful timescales and length scales are provided and are applied to observations. I recommend publication after some revision.

**Major points**

1.) A major problem seems to be the neglect of conductive heat flux, i.e. the diffusion terms which are missing in eqs. 1 and 2. This is an assumption which stems from the old Schuman paper (1929), and is subsequently applied, e.g. by Spiga and Spiga (1981). Spiga and Spiga (1981) keep the diffusion term in their non-dimensionalized equations (but neglect it later) which reads in the notation of Roy:

$$\frac{\lambda_f k}{\phi c_{pfluid}^2 \rho_{fluid}^2 v^2} \frac{\partial^2 T_f'}{\partial x'^2} \equiv C \frac{\partial^2 T_f'}{\partial x'^2} \tag{1}$$

where the primes denote non-dimensional quantities. This term and a similar term for the solid temperature needs to be added to the right hand sides of equations B1 and B2. Given some temperature variations with a certain non-dimensional wavelength $\lambda'$ one can actually check the magnitude of this term with respect to a) the advective term $\frac{\partial T_{f'}}{\partial x'}$ or b) the heat exchange term $T_f' - T_s'$ in eq. (B1). Typically $\lambda'$ is of the order 10 to 20 according to Figure A2 or A4 or shorter at earlier times. Decomposing the temperature into Fourier series we may insert typical temperature modes $T_f' = T_{f0}' sin\frac{2\pi x'}{\lambda'}$ into eq. B1 with the added diffusion term, where $T_{f0}'$ is the time-dependent amplitude of the Fourier mode. Evaluation of the resulting equation shows that neglect of the diffusive term with respect to a) the advection term or b) the heat exchange term requires

a)    $C\frac{2\pi}{\lambda'} \ll 1$         or b)    $C\left(\frac{2\pi}{\lambda'}\right)^2 \ll 1$     (2)

Inserting minimum and maximum numbers of used parameters from Table A1 of the manuscript shows that conditions (2) are severely violated. Only for the most extreme parameter combinations of $k = 10^{-5}$ W/(m³K) and $v = 100$ mm/yr the left hand sides on (2) are of order 0.1, in all other parameter combinations they are larger and violate the conditions. One can also write $C$ in terms of Peclet numbers based on the length scale $d$:

$$Pe_{\kappa_f} = \frac{v\,d}{\kappa_f} \qquad , \qquad\qquad Pe_{\kappa_{eff}} = \frac{v\,d}{\kappa_{eff}} \qquad\qquad\qquad (3)$$

where $\kappa_f$ or $\kappa_{eff}$ is the thermal diffusivity of the fluid or the effective interfacial boundary layer, respectively. Both Peclet numbers are probably very similar and can be merged to a characteristic Peclet number $Pe$. Using the equations given in the paper the relations (2) can be rewritten as

$$Pe \gg \sqrt{\frac{A(1-\phi)}{\phi}\left(\frac{2\pi}{\lambda'}\right)^n} \qquad\qquad (4)$$

where $A$ is a constant defined in the paper (2, 4, or 6) and $n = 1$ based on relation (2 a) or n = 2 for relation (2 b). From eq. (4) it follows that the neglect of the diffusion term is justified for Peclet numbers of order 1 to 10 and larger, while the Peclet numbers used in the paper are between $3 \cdot 10^{-6}$ to 0.3.

Therefore I strongly recommend to include the diffusive term into the calculations of thermal non-equilibrium and rerun the models. Note that due to numerics you probably have some numerical diffusion in your model which may be of similar order as the neglected diffusion term. Thus, you should do some resolution tests.

2.) The critical parameter is the heat transfer coefficient $k$. Already in Fig. 1 $k$ is given as proportional to $(1-\phi)/d^2$ where $\phi$ is the porosity and $d$ is the channel distance. From the physics point of view it is proportional to $(1-\phi)/(d\,\delta)$ where $\delta$ is the microscopical thermal boundary layer thickness at the solid-fluid interface (Schmeling et al., 2018). Only for long period thermal variations $\delta$ is of the order of $d$.

2.1) In Appendix C the heat transfer coefficient is discussed in more detail. There seems to be a confusion about the constant $A$ in the equation for the specific surface area $a_{sf} \approx A(1-\phi)/d$ (note that you should use the symbol "≈" or "≅" as "approximately equal" and not "~" as "proportionally" as you do correctly in the notation of $k$ in Fig. 1.). A back-of-envelope calculation results in $A = 2$ for planar channels, while for cylindrical tubes it is more complicated if written in terms of $d$ (the formula contains square roots of $\phi$). Instead, for cylindrical tubes it can be written as $a_{sf} \approx 4\phi/d_f$ where $d_f$ is the tube diameter. But correctly, it is 6 for spherical or other grains embedded in the fluid phase. In Chevalier and Schmeling (2022) we discuss some of these relations.

2.2) In eq. C1 and C2 the minus sign should be replaced by a plus (Dixon and Cresswell, 1979, eq. 29; Stuke (1948), eq 57). For $\beta$ values of 10, 8, 6 are assumed for spherical matrix grains, cylinders or slabs, respectively. Adopting Dixon and Cresswell's arguments means that short period effects (higher temporal modes as considered in Stuke, 1949) are neglected. This results from their assumption of taking Stuke's (1949) heat transfer coefficient (eq. 57 in Stuke 1949) with $\Phi = \frac{1}{\beta}$ + higher temporal orders but then neglecting these higher orders. With this assumption you get the effective conductance (your eq. C2). In my understanding, accounting for these higher orders is physically equivalent to taking the effective thermal conductivity $C_{eff}$ and then defining the effective conductance by $C_{eff}/\delta$ where $\delta$ is the microscopical thermal boundary layer thickness. By neglecting the short term higher orders one implicitly assumes that the thermal boundary layer thickness has reached the order of $d$. Only then the appropriate $k$ is given by $C_{eff} a_{sf}/d$. In other words, in your choice of $k$ you underestimate short term interfacial heat exchange. The problem with choosing $\delta$ rather than $d$ in estimating the effective conductance is that $\delta$ is time-dependent, and theoretically includes the full thermal history of the two-phase flow. In Schmeling et al (2018) we studied this effect in detail and showed that choosing $\delta = d$ describes the thermal non-equilibrium only for intermediate term evolutions, not short period thermal variations (e.g. Fig. 8 in that paper). For $\delta < d$ the heat transfer coefficient $k$ will be larger than yours, so you probably overestimate thermal non-equilibrium for short term thermal variations. My recommendation: As it is quite common in literature to use the $\delta = d$ assumption for simplicity you should keep this assumption and address and discuss this point.

3.) You don't say how you solve the equations. Please add a short section on the numerical method, grid resolution etc.

4.) The Appendices D and E contain very interesting model results. In my opinion they should be moved to the main text.

5) Discussion. In section i) you introduce the term "disequilibrium heating". This term should more rigorously be defined. In this section (e.g. Line 208) you estimate the heat budget due to disequilibrium heating by multiplying the excess infiltration temperature $\Delta T$ by $k$ to get a volumetric heat generation rate. According to eq. (2) you should use the disequilibrium temperature difference $T_f - T_s$ rather than $\Delta T$, to get an estimate of the contribution of disequilibrium heating. From your figures 2, A2 and A3 $T_f - T_s$ is of the order of $10^{-2}$ to $10^{-1}$ of $\Delta T$. These temperature differences are also consistent with the estimates of the magnitude of thermal non-equilibrium by Chevalier and Schmeling (2022) for Peclet numbers of order 0.1. Thus the estimated LAB heat budget in line 208 should be one to two orders of magnitude smaller. Yet, the advective heat transport by infiltration flow in thermal equilibrium might be larger and would be nice to be estimated here. Perhaps these estimates do fit better to the estimates of the heat released by crystallization you estimate in line 222.

6) Line 258, 260. Here you speculate about rheological weakening due to disequilibrium heating. Again, assuming 100 K as a possible temperature increase is a probably an overestimate given that the disequilibrium temperature difference $T_f - T_s$ is one to two orders of magnitude smaller than $\Delta T$. And: I have checked the activation energies and volumes of Hirth and Kohlstedt (2003) and I don't get your factors of order 1/62. I get something like 1/20 at most for constant stress, and 1/3 for constant strain rate. Given the smaller temperature difference of order 10 K reduces this effect even more to a factor 1/1.3 or something like this, which is still worthwhile to mention.

**Minor points**

7.) Line 308: you may note here that 1/z is the dimensionless channel velocity (but see also comment 13).

8.) Line 334. Are $\phi_{in}$ and $\phi_{out}$ identical to $\phi_f$ and $\phi_s$, respectively? Then you should use same symbols.

9.) Line 337. You choose $A$ and $\beta$ independently, but they are geometric parameters for spheres, tubes and spheres. Particularly $\beta$ is defined for solid spheres, cylinders and plates, while $A$ is defined for fluid tubes, etc.

10) Line 340 to 345 or section 2:  Please specify the boundary conditions more rigorously, for both $T_s$ and $T_f$ at x = 0 and at the other side of the domain. You should clearly state that $T_s'$ is also raised to 1 while you increase the influx temperature of $T_f'$ .

11.) Line 363. Delete "migration"

12.) Line 366 – 367 and line 143 – 149. The difference between the disequilibrium front velocity of Kuznetsov (1984) and your eq. 3 is puzzling and should be discussed. Is it due to different scaling? Although both, Kuznetsov's and your eq. 3 are given as dimensional equations? Or is it an effect of using perturbation theory versus full solution of the PDE's? Or is it a misprint in Kuznetsov? Anyway, how did you derive and justify eq. (3)?

13.) Fig. A2c causes confusion. From the x-label or figure caption we have

$$x'_{front} = \frac{1}{z}t'. \tag{5}$$

This implies that the disequilibrium front has the non-dimensional velocity 1/z.  But the fluid velocity may be written as

$$v_{channel} = \frac{x_f}{t} \tag{6}$$

where $x_f$ is the position of a fluid particle. If we substitute $x_f$ and $t$ using the non-dimensionalization rules one gets

$$v_{channel} = \frac{x'_f\, v_{channel}\, k_s}{t'\, k_f} = \frac{x'_f}{t'} v_{channel} z = v'_{channel}\, v_{channel} z \tag{7}$$

with $v'_{channel}$ as non-dimensional fluid velocity. After elimination of $v_{channel}$ from both sides we have

$$v'_{channel} = 1/z \qquad\qquad (8)$$

which is in contradiction to eq. (5). Can you help me (and potential readers)?

14.) Line 383. Sentence strange, probably delete one of the "is" or insert "which"

15.) Line 390 – 391. Which "blue lines"? Do you mean the dashed lines or the double arrows?

16.) Line 391: "wavelength" probably to be replaced by "period"

17.) Line 149. I don't see the strong function of $k$ in Fig. A4.

18.) Line 150 – 159. You clearly describe the exponential decay of disequilibrium. Could you elaborate a bit on the decay rate for the step function case?

19.) Line 163: delete one of the parantheses ")" in the first *tanh* term

20.) Conclusion: Here I suggest to repeat the meaning of the abbreviations CLM, TRZ again

**References**

Chevalier, L. and Schmeling, H.: Thermal non-equilibrium of porous flow in a resting matrix applicable to melt migration: a parametric study, Solid Earth Discuss. [preprint], https://doi.org/10.5194/se-2021-149, in review, 2021.

Dixon, A. G. and Cresswell, D. L.: Theoretical prediction of effective heat transfer parameters in packed beds, AIChE Journal, 25, 1979.

Hirth, G. and Kohlstedt, D. L.: Rheology of the Upper Mantle and the Mantle Wedge: A View from the Experimentalists, Geophysical Monograph Series, 138, 83–105, 2003.

Kuznetsov, A. V.: An investigation of a wave of temperature difference between solid and fluid phases in a porous packed bed, International Journal of Heat and Mass Transfer, 37, 3030–3033, 1994.

Schmeling, H., Marquart, G., and Grebe, M.: A porous flow approach to model thermal non-equilibrium applicable to melt migration. Geophys. J. Int., Volume 212, Issue 1, 1 January 2018, Pages 119–138, doi: 10.1093/gji/ggx406, 2018

Schumann, T. E.: Heat transfer: a liquid flowing through a porous prism, Journal of the Franklin Institute, 208, 405–416, 1929.

Spiga, G. and Spiga, M.: A rigorous solution to a heat transfer two-phase model in porous media and packed beds, International Journal of Heat and Mass Transfer, 24, 355–364, 1981.

Stuke, B., Berechnung des Wärmeaustausches in Regeneratoren mit zylindrischen und kugelförmigen Füllmaterial, Angewandte Chemie, B20,262, 1948.

---

## Referee Comment (RC2)

**Review of "Assessing the role of thermal disequilibrium in the evolution of the lithosphere-asthenosphere boundary: An idealized model of heat exchange during channelized melt-transport"**

**I. GENERAL COMMENTS**

This paper investigates the potential for channelized melt transport into the base of the thermal lithosphere to supply an elevated localized heat flux. A simple modelling approach is used with a series of idealized forcing scenarios. The results include calculations of the scale of the thermal reworking zone and estimates of the overall heat supply. The modelling approach is heavily idealized and so is subject to significant limitations. The writing of the paper was hard to follow in parts. This could be improved by restructuring as a coherent whole without the back-and-forth use of appendices to develop both theory and results. However, the topic is interesting and the modelling is a useful starting point that makes a good contribution to analyzing the problem. Overall, I think the paper should be *accepted subject to minor revisions*.

**II. SPECIFIC COMMENTS**

1. **Explanation of model, especially relating to the heat transfer and channel spacing:** In the specific comments section, I give several suggestions for how I thought the model and results could be explained more clearly. This includes some suggested revisions to the notation. My main concern in this area relates to the modelling of the heat transfer process. Physically, I would think that macroscale heat transfer results from microscale diffusion (i.e. thermal conduction), but the paper states that axial conduction is neglected. But looking at the appendix (C2), $k$ is proportional to an effective thermal conductivity divided by the square of some length scale ($d$ in the equation). The correct choice/s of length scale is the crucial issue (the square is clear from dimensional grounds). The authors say that $d$ is the channel spacing, but elsewhere (L316) say that $d$ is the particle diameter. These are obviously very different. So the relevant length scale needs much better justification and the role of conduction (equivalently diffusion) in the model should be clearer.

2. **Simplifications in modelling approach:** There are numerous simplifications inherent in the modelling approach. These are generally mentioned in the text but I felt the paper would benefit from

more analysis of the relative importance of the various simplifications made. Two related simplifications that seem especially important to me relate to the parameters $\phi$ (fraction occupied by channel), the make up of the channels, and thermal (and perhaps chemical feedbacks). It is not entirely clear whether the channels are envisaged as purely liquid, narrow dikes surrounded by entirely solid rock or much wider bodies of partially molten rock, where a channel is distinguished as having a higher melt fraction. In either case, it is clear that the properties of these channels are in practice determined that the operative dynamics and it is a large simplification to just impose them. There must also be feedbacks between any thermal reworking process and the channels themselves but this can't be investigated within this type of model, as the channel properties are just imposed.

3. **Paper structure:** Significant aspects of the paper were hard to follow. I was less concerned about appendix A (but also don't see why a few short paragraphs couldn't be included in the introduction). Appendix C develops substantial aspects of the model (including aspects novel or specific to this study) to such an extent that the description in the main text relies heavily on material in the appendix (e.g. the discussion of $k$, $k_f$ and $k_s$, which are crucial to the paper). Appendix B is rather more technical, but the meaning of symbols developed there is relied on elsewhere. So it should either be incorporated into the main text, or care should be taken such that all notation is properly defined in the main text at least. Appendix D and especially appendix E, given that it is perhaps the most 'realistic' scenario considered, also belong in the main results section. The summary given relies on notation developed in the appendices as well as figures only reported in the appendices. For this style of journal, the back-and-forth between main text and appendices is hard to justify.

**III. TECHNICAL CORRECTIONS**

4. **L33–45 or final paragraph of introduction:** Consider referring to body of work relating to thinning of the thermal lithosphere in arc settings (e.g. England and Katz, 2010, `https://doi.org/10.1038/nature09417`, Perrin et al., 2016, `https://agupubs.onlinelibrary.wiley.com/doi/10.1002/2016GC006527` and Rees Jones et al., 2018, `https://doi.org/10.1016/j.epsl.2017.10.015`.)

5. **L51-54:** this is a very significant simplification as it precludes any feedbacks between the channels and the process(es) that create them.

6. **L58:** '$v$ is transport velocity' needs a bit more explanation (transport velocity of what?). Also I assume from the equations that the solid is not moving but this could be stated more clearly in the text. I don't really understand why you introduce a new symbol $v_{\text{channel}}$ when it seems to be the same as $v$. The cartoon sketch in figure 1 is also a bit unclear as to whether $v$ is the fluid velocity within the narrow channels in the zoomed in circles or some kind of average?

7. **Eqs. 1–2:** This way of defining $k_f$ and $k_s$ could be clearer. The notation is also potentially confusing as $k$ has different units from $k_s$ and $k_f$. Suggest changing one of the symbols.

8. **Figure 1:** These time-dependent forcings have very different total energy inputs which could be emphasized a bit more, perhaps.

9. **L87 & L109:** 'across channel walls' sounded a bit strange because the fluid flow seemed to be vertical so there wouldn't be much flow across channel walls, since the walls in the sketch are also near vertical.

10. **L104-112:** consider phrasing this discussion in terms of a Peclet number.

11. **L136:** Think 'duration' was intended rather than 'amplitude.'

12. **L138–:** Think that this section would be easier to understand if text from appendix (and especially figures) was included in the main text.

13. **Figure 2:** This is a useful figure. But I think plots against $x$ at a series of $t$ values are also useful complementary way to show the same data.

14. **L174:** 10 m.

15. **Figure 3:** Consider plotting agains the theoretical scaling to collapse all the data on a single line.

16. **Figure 4 & L231:** I wondered if this velocity range was rather low, for example when compared to typical asthenospheric melt velocities which might be an order of magnitude larger.

17. **L203–224:** Perhaps it would make more sense to consider the overall LAB heat budget rather than one component.

18. **L305:** $z$ is an odd choice of symbol (looks more like a vertical coordinate) and could be defined more clearly.

19. **L310:** Might benefit from a brief discussion of the numerical methods used.

20. **L311 & 316:** $d$ appears to be used for two different quantities

21. **eqs. C1 & C2:** check whether the minus sign is correct. This looks like it should be related to the harmonic mean of two conductivities (it would be with a plus sign). And the equations would be problematic if the term in square brackets were zero.

22. **L325:** Not sure where this range came from originally but I don't think it would be appropriate if the model is intended to be of a porous flow, it sounds more like a pipe flow argument.

23. **Figure A3:** Could benefit from better formatting to match the standard of the other figures.

---

## Author Comment (AC1)

**Overall:**

*"In this manuscript the thermal effect of melt infiltration into the base of the continental lithosphere is studied focusing on the thermal disequilibrium between melt and ambient rock. While thermal disequilibrium in porous flow is well studied in more technical literature, only a few papers quantitatively addressed this effect in the recent geoscience literature. Therefore this paper is timely and new. It is shown that indeed thermal disequilibrium may be important under certain circumstances near the lithosphere asthenosphere boundary explaining some observational data. Useful timescales and length scales are provided and are applied to observations. I recommend publication after some revision."*

**Thank you very much for your thoughtful and detailed comments. I am very grateful for the time and effort you have devoted to this review—your comments are invaluable and will greatly improve this paper.**

I want to post an **initial response**, below, in which I address each of your comments and sketch out an outline of how I would address these in a potential revised manuscript, should the editor allow a revision. I understand that I am to wait until the editor's decision is made and, **if I am allowed to submit a revised manuscript, I plan to upload a final, more complete version of my responses to the comments, referring to line numbers in the revised manuscript.**

**Major comments**

| | |
|---|---|
| 1. A major problem seems to be the neglect of conductive heat flux, i.e. the diffusion terms which are missing in eqs. 1 and 2.

… | Yes, this is an important critique and you are correct that the model assumptions are invalid when the heat transfer coefficient is too large. I had tested inequalities (1) and (2) you derived using Fourier modes, but for an earlier set of models with smaller heat transfer coefficient, $k$ |
| ... Therefore I strongly recommend to include the diffusive term into the calculations of thermal non-equilibrium and rerun the models. | **I am redoing these calculations including the diffusion terms to test the robustness of my interpretations of the TRZ and the overall heat LAB budget for large Peclet numbers. I am aware that the conclusions of the current manuscript may undergo modification.** The goal of this paper is to set limits on the importance of disequilibrium heat exchange within the lowermost continental lithosphere, and this should still be possible with the suggested change to the model. |
| Note that due to numerics you probably have some numerical diffusion in your model which may be of similar order as the neglected diffusion term. Thus, you should do some resolution tests. | To avoid this, I choose dt to satisfy the CFL condition. I plan on describing my numerical methods in more detail in a short section in the revised paper. Currently I am using an explicit leapfrog method with center-time and $2^{nd}$ order upwind finite difference scheme; see response comment #3 |
| 2. The critical parameter is the heat transfer coefficient $k$. Already in Fig. 1 $k$ _is given as proportional to $(1-\phi)/d^2$ where $\phi$ _is the porosity and $d$ is the channel distance. From the physics point of view it is proportional to $(1-\phi)/(d\,\delta)$ where $\delta$ is the microscopical thermal boundary layer thickness at | This is a good point. For the timescales of the driving term here, we consider durations that are longer than $1/k_s$, a nominal thermal response timescale for the solid. I plan to touch on this, and the relationship to the microscopic treatment in your 2018 paper, in the revised paper. |

| | |
|---|---|
| the solid-fluid interface (Schmeling et al., 2018). Only for long period thermal variations $\delta$ is of the order of $d$. | |
| 2.1) In Appendix C the heat transfer coefficient is discussed in more detail. There seems to be a confusion about the constant $A$ in the equation for the specific surface area $a_{sf} \approx A(1-\phi)/d$ | |
| (note that you should use the symbol "$\approx$" or "$\cong$" as "approximately equal" and not "$\sim\_$" as "proportionally" as you do correctly in the notation of $k\_$in Fig. 1.). | OK, yes will fix this. |
| A back-of-envelope calculation results in $A=2$ or planar channels, while for cylindrical tubes it is more complicated if written in terms of $d$ (the formula contains square roots of $\phi$). Instead, for cylindrical tubes it can be written as $a_{sf}\approx4\_\phi/\_d_f\_$where $d_f$ is the tube diameter. But correctly, it is 6 for spherical or other grains embedded in the fluid phase. In Chevalier and Schmeling (2022) we discuss some of these relations. | I shall cite this paper in connection to the discussion of reasonable numbers for A |
| 2.2) In eq. C1 and C2 the minus sign should be replaced by a plus (Dixon and Cresswell, 1979, eq. 29; Stuke (1948), eq 57). For $\beta\_$values of 10, 8, 6 are assumed for spherical matrix grains, cylinders or slabs, respectively. | Yes, thank you! – this is a typo in both C1 and C2 |
| Adopting Dixon and Cresswell's arguments means that short period effects (higher temporal modes as considered in Stuke, 1949) are neglected. This results from their assumption of taking Stuke's (1949) heat transfer coefficient (eq. 57 in Stuke 1949) with $\Phi=1/\beta+$ higher temporal orders but then neglecting these higher orders. With this assumption you get the effective conductance (your eq. C2). In my understanding, accounting for these higher orders is physically equivalent to taking the effective thermal conductivity $C_{eff}$ and then defining the effective conductance by $C_{eff}/\delta$ where $\delta$is the microscopical thermal boundary layer thickness. By neglecting the short term higher orders one implicitly assumes that the thermal boundary layer thickness has reached the order of $d$. Only then the appropriate $k\_$is given by $C_{eff}a_{sf}/d$. **In other words, in your choice of $k$ you underestimate short term interfacial heat exchange**. The problem with choosing $\delta$ rather than $d$ in estimating the effective conductance is that $\delta$ is time-dependent, and theoretically includes the full thermal history of the two-phase flow. In Schmeling et al | Yes, I need to address this as a limitation of this way or estimating the effective conductance.  I will essentially discuss the caveat that this underestimates the heat transfer coefficient for short term variations, limiting its applicability to thermal driving terms that vary over timescales that are "long" per your 2018 paper. |

| | |
|---|---|
| (2018) we studied this effect in detail and showed that choosing $\delta=d$ describes the thermal non-equilibrium only for intermediate term evolutions, not short period thermal variations (e.g. Fig. 8 in that paper). For $\delta<d$ the heat transfer coefficient $k$ will be larger than yours, so you probably overestimate thermal non-equilibrium for short term thermal variations. My recommendation: As it is quite common in literature to use the $\delta=d$ assumption for simplicity you should keep this assumption and address and discuss this point. | |
| 3.) You don't say how you solve the equations. Please add a short section on the numerical method, grid resolution etc. | I plan to add a section on my numerical solution methods. The code is simple, written in Matlab and uses an explicit leapfrog method with center-time and $2^{nd}$ order upwind finite difference scheme The calculation is carried out on a 1D domain (N=5000 elements in most models) with dt chosen to obey the CFL condition (so it depends on $z$). To test the accuracy of the solution (e.g., when comparing to Schumann's and Kenyon's analytic solutions) I considered grid resolution tests and chose a model size that is appropriate. This optimization will likely need to be revised when the additional diffusion terms are included. I shall include a discussion of this in the revised manuscript. |
| 4.) The Appendices D and E contain very interesting model results. In my opinion they should be moved to the main text. | Yes, both reviewers point out the problematic flow between the main text and the Appendices. I shall move these sections into the main text to improve the flow. |
| 5) Discussion. In section i) you introduce the term "disequilibrium heating". This term should more rigorously be defined. In this section (e.g. Line 208) you estimate the heat budget due to disequilibrium heating by multiplying the excess infiltration temperature $\Delta T$ _by $k$ _to get a volumetric heat generation rate. According to eq. (2) you should use the disequilibrium temperature difference $T_f - T_s$ _rather than $\Delta T$,... | Agreed. I will use $(T_f - T_s)_{max}$ instead of $\Delta T$ and explain this in the context of the implied disequilibrium heat exchange. |
| 6) Line 258, 260. Here you speculate about rheological weakening due to disequilibrium heating. Again, assuming 100 K as a possible temperature increase is a probably an overestimate given that the disequilibrium temperature difference $T_f - T_s$ _is one to two orders of magnitude smaller than $\Delta T$. And: I have checked the activation energies and volumes of Hirth and Kohlstedt (2003) and I don't get your factors of order 1/62. I get something like 1/20 at most for constant stress, and 1/3 for constant strain rate. Given | Yes, again I should more properly use $(T_f - T_s)_{max}$

 OK, will check again |

| | |
|---|---|
| the smaller temperature difference of order 10 K reduces this effect even more to a factor 1/1.3 or something like this, which is still worthwhile to mention. | Yes. |

Minor points:

| | |
|---|---|
| 7.) Line 308: you may note here that $1/z$ is the dimensionless channel velocity (but see also comment 13). | Yes, $1/z$ is indeed the dimensionless channel velocity – see also response to #13 |
| 8.) Line 334. Are $\phi_{in}$ and $\phi_{out}$ identical to $\phi_f$ and $\phi_s$, respectively? Then you should use same symbols. | These are only identical if we take the 'end member' case where the channels are pure fluid and walls are solid. I was trying to say here that this does not need to be the case, but will clarify |
| 9.) Line 337. You choose $A$ and $\beta$ independently, but they are geometric parameters for spheres, tubes and spheres. Particularly $\beta$ is defined for solid spheres, cylinders and plates, while $A$ is defined for fluid tubes, etc. | Yes, you are correct. I do this to investigate how the estimates of $k$ are affected by a range in A and $\beta$ – will clarify in revision |
| 10) Line 340 to 345 or section 2: Please specify the boundary conditions more rigorously, for both $T_s$ and $T_f$ at $x = 0$ and at the other side of the domain. You should clearly state that $T_s'$ is also raised to 1 while you increase the influx temperature of $T_f'$. | Yes, your point here and #13 below clearly show that the boundary and initial conditions need to be clarified. |
| 11.) Line 363. Delete "migration" | OK |
| 12.) Line 366 – 367 and line 143 – 149. The difference between the disequilibrium front velocity of Kuznetsov (1984) and your eq. 3 is puzzling and should be discussed. Is it due to different scaling? Although both, Kuznetsov's and your eq. 3 are given as dimensional equations? Or is it an effect of using perturbation theory versus full solution of the PDE's? Or is it a misprint in Kuznetsov? Anyway, how did you derive and justify eq. (3)? | I **think** that this arises because he is using an analytic (perturbative) approach whereas I am solving the full system. I will explain this better, but using the velocity estimate in Kuznetsov as a starting point (a first guess), I find (empirically) that Eqn 3 best describes the velocity in my models. I will explain this in revision, but I am afraid that I don't have a derivation. |
| 13.) Fig. A2c causes confusion. From the x-label or figure caption we have $x'_{front}=(1/z)t'$. (5) This implies that the disequilibrium front has the non-dimensional velocity $1/z$. But the fluid velocity may be written as $v_{channel}=x_f t$ (6) where $x_f$ is the position of a fluid particle. If we substitute $x_f$ and $t$ using the non-dimensionalization rules one gets $v_{channel}=\_x_{f'}\_v_{channel}\_k_s t'\_k_f=\_x_{f'}\_t'v_{channel}z=\_v'_{channel}\_v_{channel}z\_$ (7) | The confusion lies here: the initial conditions needed to be better described (currently in Appendix D): For $t<0$, there is material flowing in the channels, at $v_{channel}$, but the channels are at the same initial temperature as the walls, $T_0 = T_s = T_f$. At $t=0$, the temperature of the material entering the channels at $x=0$ is perturbed. So, what I mean by $x_{front}$ **is not the location of the disequilibrium front**. Instead it is the location of the in-channel material that entered the domain at $x=0$ at $t=0$. This material moves at the speed $v_{channel}$ relative to the walls and at a later time t, it is located at $x=x_{front} = v_{channel}$ t. Therefore, there is no |

| | |
|---|---|
| with $v'_{channel}$ as non-dimensional fluid velocity. After elimination of $v_{channel}$ from both sides we have $v'_{channel} = \_1\_/\_z\_$ (8) which is in contradiction to eq. (5). Can you help me (and potential readers)? | contradiction, and 1/z is indeed the dimensionless velocity of material in the channels. I think the word "front" here is causing the confusion. In the revised paper, I will change this to something like $x_{pert}$ to more clearly indicate that this is the location of the material that entered with a perturbed temperature. |
| 14.) Line 383. Sentence strange, probably delete one of the "is" or insert "which" | OK |
| 15.) Line 390 – 391. Which "blue lines"? Do you mean the dashed lines or the double arrows? | Thanks for catching a typo: these should refer to the double arrows (they used to be blue in a previous version) |
| 16.) Line 391: "wavelength" probably to be replaced by "period" | No, it is wavelength: I mean the peak-to-peak distance (at a fixed time) |
| 17.) Line 149. I don't see the strong function of $k$ in Fig. A4. | Ok, it should refer to the functional dependence on $d$, which in turn strongly controls $k$ |
| 18.) Line 150 – 159. You clearly describe the exponential decay of disequilibrium. Could you elaborate a bit on the decay rate for the step function case? | The sentence is confusing… the exponential decay I am referring to is a spatial decay as a function of distance along the transport direction. So, the successive peaks in the blue curve on figure A2b should go down by roughly the same factor as they are their distance apart is roughly the same. I will check this and report it quantitatively in the revision. |
| 19.) Line 163: delete one of the parantheses ")" in the first *tanh* term | Yes |
| 20.) Conclusion: Here I suggest to repeat the meaning of the abbreviations CLM, TRZ again | OK |

---

## Author Comment (AC2)

Initial Response to RC2                                                                    4 April, 2022
Mousumi Roy

**Overall:**

''This paper investigates the potential for channelized melt transport into the base of the thermal lithosphere to supply an elevated localized heat flux. A simple modelling approach is used with a series of idealized forcing scenarios. The results include calculations of the scale of the thermal reworking zone and estimates of the overall heat supply. The modelling approach is heavily idealized and so is subject to significant limitations. The writing of the paper was hard to follow in parts. This could be improved by restructuring as a coherent whole without the back-and-forth use of appendices to develop both theory and results. However, the topic is interesting and the modelling is a useful starting point that makes a good contribution to analyzing the problem. Overall, I think the paper should be *accepted subject to minor revisions."*

**Thank you very much for your review. I appreciate the thoughtful and constructive comments you have provided; I gratefully acknowledge they will improve the paper and increase its impact and readability.**

I want to post an **initial response**, below, in which I address each of your comments and sketch out an outline of how I would address these in a potential revised manuscript, should the editor allow a revision. I understand that I am to wait until the editor's decision is made and, **if I am allowed to submit a revised manuscript, I plan to upload a final, more complete version of my responses to the comments, referring to line numbers in the revised manuscript.**

**General comments**

| | |
|---|---|
| 1. Explanation of model, especially relating to the heat transfer and channel spacing: In the specific comments section, I give several suggestions for how I thought the model and results could be explained more clearly. This includes some suggested revisions to the notation. | Thank you for this comment. As R1 has pointed out also, the grounds for neglecting axial conduction are not valid for some of the models I consider. |
| My main concern in this area relates to the modelling of the heat transfer process. Physically, I would think that macroscale heat transfer results from microscale diffusion (i.e. thermal conduction), but the paper states that axial conduction is neglected. But looking at the appendix (C2), k is proportional to an effective thermal conductivity divided by the square of some length scale (d in the equation). The correct choice/s of length scale is the crucial issue (the square is clear from dimensional grounds). The authors say that d is the channel spacing, but elsewhere (L316) say that d is the particle diameter. These are obviously very different. So the relevant length scale needs much better justification and the role of conduction (equivalently diffusion) in the model should be clearer. | **I am redoing these calculations including the diffusion terms to test the robustness of my interpretations of the TRZ and the overall heat LAB budget for large Peclet numbers. I am aware that the conclusions of the current manuscript may undergo modification.** The goal of this paper is to set limits on the importance of disequilibrium heat exchange within the lowermost continental lithosphere, and this should still be possible with the suggested change to the model. |
| 2. Simplifications in modelling approach: There are numerous simplifications inherent in the modelling approach. These are generally mentioned in the text but I felt the paper would benefit from | Yes, related to 1 above, I plan to also discuss the relative importance of axial conduction terms in the equations and address how including these affects the overall story of the paper. |

| more analysis of the relative importance of the various simplifications made. Two related simplifications that seem especially important to me relate to the parameters φ (fraction occupied by channel), the make up of the channels, and thermal (and perhaps chemical feedbacks). | Chemical exchange is ignored here entirely and this is indeed a limitation, but it will be more clearly stated in the revised paper. |
|---|---|
| It is not entirely clear whether the channels are envisaged as purely liquid, narrow dikes surrounded by entirely solid rock or much wider bodies of partially molten rock, where a channel is distinguished as having a higher melt fraction. In either case, it is clear that the properties of these channels are in practice determined that the operative dynamics and it is a large simplification to just impose them. There must also be feedbacks between any thermal reworking process and the channels themselves but this can't be investigated within this type of model, as the channel properties are just imposed. | As I mention in lines 330-339 in the current manuscript, the channels may be a high-porosity region within a lower-porosity surrounding region. To explore the 'end-member' upper limit to the disequilibrium heat exchange, I consider the case where the channels are purely liquid and walls are solid. This needs to be clarified in the main text and not in an appendix (see response to #3 below).

Yes, there are no transport and channelization feedbacks that can be explored in this limited approach (but see lines 50-55; 98-100 in current manuscript).  I plan to make this clearer in my revision. |
| 3. Paper structure: Significant aspects of the paper were hard to follow. I was less concerned about appendix A (but also don't see why a few short paragraphs couldn't be included in the introduction). Appendix C develops substantial aspects of the model (including aspects novel or specific to this study) to such an extent that the description in the main text relies heavily on material in the appendix (e.g. the discussion of k, $k_f$ and $k_s$, which are crucial to the paper). Appendix B is rather more technical, but the meaning of symbols developed there is relied on elsewhere. So it should eitherbe incorporated into the main text, or care should be taken such that all notation is properly defined in the main text at least. | OK, thank you for this comment.  I feel your suggestions will greatly improve the flow of the paper and strengthen its impact.  This is also in line with R1's comments on the organization of the Appendices and the material in the text. |
| Appendix D and especially appendix E, given that it is perhaps the most 'realistic' scenario considered, also belong in the main results section. The summary given relies on notation developed in the appendices as well as figures only reported in the appendices. For this style of journal, the back-and-forth between main text and appendices is hard to justify. | Yes. |

Technical comments:

| | |
|---|---|
| 4. L33–45 or final paragraph of introduction: Consider referring to body of work relating to thinning of the thermal lithosphere in arc settings (e.g. England and Katz, 2010, https://doi.org/10.1038/nature09417, Perrin et al., 2016, https://agupubs.onlinelibrary.wiley.com/doi/10.1002/2016GC006527 and Rees Jones et al., 2018, https://doi.org/10.1016/j.epsl.2017.10.015.) | Yes, I shall include some of these papers. |
| 5. L51-54: this is a very significant simplification as it precludes any feedbacks between the channels and the process(es) that create them. | Yes. It is highlighted in the beginning of the Discussion also, currently lines 176-186. |
| 6. L58: 'v is transport velocity' needs a bit more explanation (transport velocity of what?). Also I assume from the equations that the solid is not moving but this could be stated more clearly in the text. I don't really understand why you introduce a new symbol $v_{channel}$ when it seems to be the same as v. The cartoon sketch in figure 1 is also a bit unclear as to whether v is the fluid velocity within the narrow channels in the zoomed in circles or some kind of average? | OK, yes, this is a typo. The velocity should be $v_{channel}$ everywhere. I plan to retain the 'channel' to specify that this is the average rate of relative motion of material within and outside channels.

I need to clarify this as an average rate |
| 7. Eqs. 1–2: This way of defining $k_f$ and $k_s$ could be clearer. The notation is also potentially confusing as k has different units from $k_s$ and $k_f$. Suggest changing one of the symbols. | OK, your point about the units being different is very good and I will change the notation so *k* is only used as the effective heat transfer coefficient. |
| 8. Figure 1: These time-dependent forcings have very different total energy inputs which could be emphasized a bit more, perhaps. | OK. |
| 9. L87 & L109: 'across channel walls' sounded a bit strange because the fluid flow seemed to be vertical so there wouldn't be much flow across channel walls, since the walls in the sketch are also near vertical. | I shall clarify; I mean relative motion between material inside and outside channels. |
| 10. L104-112: consider phrasing this discussion in terms of a Peclet number. | Yes, this was also brought up by R1 and I will include this in the revision |
| 11. L136: Think 'duration' was intended rather than 'amplitude.' | Yes, you are correct. |
| 12. L138–: Think that this section would be easier to understand if text from appendix (and especially figures) was included in the main text. | OK, agreed. |
| 13. Figure 2: This is a useful figure. But I think plots against x at a series of t values are also useful complementary way to show the same data. | Yes, I will include this in a revision |

| | |
|---|---|
| 14. L174: 10 m. | Yes, I need the units |
| 15. Figure 3: Consider plotting agains the theoretical scaling to collapse all the data on a single line. | I thought about this, but decided against it because the effect of $d$ on $\delta$ is important to show visually. |
| 16. Figure 4 & L231: I wondered if this velocity range was rather low, for example when compared to typical asthenospheric melt velocities which might be an order of magnitude larger. | OK. I plan to revisit this after correcting the calculations to include the diffusion terms |
| 17. L203–224: Perhaps it would make more sense to consider the overall LAB heat budget rather than one component. | Agreed. But the stated goals of this work are to place limits on this one process, namely disequilibrium heat exchange. I can state this explicitly here again. |
| 18. L305: z is an odd choice of symbol (looks more like a vertical coordinate) and could be defined more clearly. | OK. This is in keeping with some of the previous literature I cite. I can see how this would be confusing though and will think about how I can clarify this. |
| 19. L310: Might benefit from a brief discussion of the numerical methods used. | Yes, agreed. I plan to add a short section on this and on grid resolution tests. |
| 20. L311 & 316: d appears to be used for two different quantities | Yes, this is confusing! It will be corrected by using a different symbol for particle diameter. |
| 21. eqs. C1 & C2: check whether the minus sign is correct. This looks like it should be related to the harmonic mean of two conductivities (it would be with a plus sign). And the equations would be problematic if the term in square brackets were zero. | Yes, absolutely; As also pointed out by R1, another typo – thank you!! |
| 22. L325: Not sure where this range came from originally but I don't think it would be appropriate if the model is intended to be of a porous flow, it sounds more like a pipe flow argument. | OK. I agree that it is tricky in a 'coarse grained' model such as this to connect to microscopic geometry. My intention here is to illustrate what reasonable numbers might be for A and $\beta$… as Reviewer 1 suggested, I can connect to some previous work to motivate this better. |
| 23. Figure A3: Could benefit from better formatting to match the standard of the other figures | OK – I am guessing you mean panels (c) and (d) in particular. I will fix this. |

---

## Author Response (AR1)

**Overall:**

*"In this manuscript the thermal effect of melt infiltration into the base of the continental lithosphere is studied focusing on the thermal disequilibrium between melt and ambient rock. While thermal disequilibrium in porous flow is well studied in more technical literature, only a few papers quantitatively addressed this effect in the recent geoscience literature. Therefore this paper is timely and new. It is shown that indeed thermal disequilibrium may be important under certain circumstances near the lithosphere asthenosphere boundary explaining some observational data. Useful timescales and length scales are provided and are applied to observations. I recommend publication after some revision."*

**Thank you very much for your thoughtful and detailed comments.  I am very grateful for the time and effort you have devoted to this review—your comments are invaluable and will greatly improve this paper.**

Here is a brief summary of major changes made to the paper:
1. Calculations repeated with a system of equations including axial conduction terms – also a comparison of solutions with and without diffusion in the step-function perturbation case
2. Material in previously in Appendices is now within the flow of the main text.
3. Introduction and discussion of possible definitions of an effective Pe number – revision of parameter range for channel spacing *d* to ensure Pe > 10 here
4. Modifications to results:
   a. Reduction in the estimated contribution of disequilibrium heating as defined here to the LAB heat budget
   b. Scaling exponent for TRZ width δ as a function of characteristic perturbation timescale is between n=1 to 2, whereas it was closer to 2 without diffusion term
5. A short section describing numerical methods
6. Clarification of notations for heat transfer coefficient (*K* not k; $K_s$ and $K_f$); dimensionless velocity is now ζ not *z* (which seemed like a coordinate)

Below I address each of your comments and point to revisions (line numbers) in the revised manuscript (changes marked in red on revised PDF).

**Major comments**

| 1. A major problem seems to be the neglect of conductive heat flux, i.e. the diffusion terms which are missing in eqs. 1 and 2. … | Yes, this is an important critique and you are correct that the model assumptions are invalid when the heat transfer coefficient is too large. I had tested inequalities (1) and (2) you derived using Fourier modes, but for an earlier set of models with smaller heat transfer coefficient, *K* |
|---|---|
| .. From eq. (4) it follows that the neglect of the diffusion term is justified for Peclet numbers of order 1 to 10 and larger, while the Peclet numbers used in the paper are between $3 \cdot 10_{-6}$ to 0.3. | Yes, you are correct.  I do restrict the interpretations in the revised paper to models with Pe > 10 now.  In Figure 2, which explores factors that control the heat transfer coefficient, *K*, I keep a broader range of channel spacing (d) values, but clearly state that we shall restrict our models to large enough channel spacings where advection dominates over diffusion. |
| ... Therefore I strongly recommend to include the diffusive term into the calculations of thermal non-equilibrium and rerun the models. | Yes, even though the key conclusions and the figures are all now based on calculations for Pe > 10, I have redone all the calculations now with the axial conduction terms included.  Primarily, I wanted to test the robustness of my interpretations of the TRZ and the overall heat LAB budget for large Peclet numbers |

and to compare the results with and without conduction. The conclusions of the paper have not undergone significant modification and the overall findings of the previous version of this study hold. In summary, key results that remain unchanged are:

1. A limit is set on the importance of disequilibrium heat exchange within the lowermost continental lithosphere, but the heat budget is now revised to be lower than previously, comparable to that estimated from the deposition of latent heat (lines 354-360; 374-377).

2. Documenting the rate at which the zone of disequilibrium heat exchange progresses inward into the domain from the inlet.

3. Showing the likelihood that a TRZ forms at the base of the CLM for geologically-reasonable parameters and that the width of the TRZ is proportional to the characteristic thermal perturbation timescale.

Note that both terms on the right hand side of the governing equations 1&2, the $(T_f - T_s)$ linear driving and the axial conduction terms, will be responsible for the broadening of any initially-sharp thermal pulse or the shallowing of an initially-steep thermal gradients. Even in the absence of diffusion, initially steep gradients will shallow as the channel material cools while the surroundings heat in the models. This is now shown in Figure 3 in c & d and discussed in lines 255-260.

| | |
|---|---|
| Note that due to numerics you probably have some numerical diffusion in your model which may be of similar order as the neglected diffusion term. Thus, you should do some resolution tests. | I assume that by numerical diffusion you are referring to the dispersion that gives rise to instability in explicit methods, determined by the Courant-Friedrichs-Lewy or CFL condition. I have seen the term numerical diffusion in the context of broadening of initially-sharp boundaries that arises in a 2D or 3D discretization of the advection-diffusion equation (where flow is not purely along x, y, or z). I now describe my numerical methods in more detail in a short section in the revised paper. Lines 220-227 |
| 2. The critical parameter is the heat transfer coefficient $k$. Already in Fig. 1 $k$ _is given as proportional to $(1-\phi)/d^2$ where $\phi$ _is the porosity and $d$ is the channel distance. From the physics point of view it is proportional to $(1-\phi)/(d\,\delta)$ where $\delta$ is | This is a good point. For the timescales of the driving term here, we consider durations that are longer than $1/K_s$, a nominal thermal response timescale for the solid. I touch on this, and the relationship to the |

the microscopical thermal boundary layer thickness at the solid-fluid interface (Schmeling et al., 2018). Only for long period thermal variations $\delta$ is of the order of $d$.

| microscopic treatment in your 2018 paper, in the revised paper. Lines 174-183.

2.1) In Appendix C the heat transfer coefficient is discussed in more detail. There seems to be a confusion about the constant $A$ in the equation for the specific surface area $a_{sf} \approx A(1-\phi)/d$

(note that you should use the symbol "$\approx$" or "$\cong$" as "approximately equal" and not "$\sim$_" as "proportionally" as you do correctly in the notation of $k$ _in Fig. 1.).

| OK, yes will fix this.

A back-of-envelope calculation results in $A=2$ or planar channels, while for cylindrical tubes it is more complicated if written in terms of $d$ (the formula contains square roots of $\phi$). Instead, for cylindrical tubes it can be written as $a_{sf} \approx 4\_\phi/\_d_f\_$ where $d_f$ is the tube diameter. But correctly, it is 6 for spherical or other grains embedded in the fluid phase. In Chevalier and Schmeling (2022) we discuss some of these relations.

| I now cite this paper in connection to the discussion of reasonable numbers for A; Lines 152-153; 175-177

2.2) In eq. C1 and C2 the minus sign should be replaced by a plus (Dixon and Cresswell, 1979, eq. 29; Stuke (1948), eq 57). For $\beta$ _values of 10, 8, 6 are assumed for spherical matrix grains, cylinders or slabs, respectively.

| Yes, thank you! – this is a typo in both C1 and C2

Adopting Dixon and Cresswell's arguments means that short period effects (higher temporal modes as considered in Stuke, 1949) are neglected. This results from their assumption of taking Stuke's (1949) heat transfer coefficient (eq. 57 in Stuke 1949) with $\Phi = 1/\beta +$ higher temporal orders but then neglecting these higher orders. With this assumption you get the effective conductance (your eq. C2). In my understanding, accounting for these higher orders is physically equivalent to taking the effective thermal conductivity $C_{eff}$ and then defining the effective conductance by $C_{eff}/\delta$ where $\delta$ is the microscopical thermal boundary layer thickness. By neglecting the short term higher orders one implicitly assumes that the thermal boundary layer thickness has reached the order of $d$. Only then the appropriate $k$ _is given by $C_{eff}a_{sf}/d$. **In other words, in your choice of $k$ you underestimate short term interfacial heat exchange**. The problem with choosing $\delta$ rather than $d$ in estimating the effective conductance is that $\delta$ is time-dependent, and theoretically includes the full thermal

| Yes, I need to address this as a limitation of this way or estimating the effective conductance.  I discuss the caveat that this underestimates the heat transfer coefficient for short term variations, limiting its applicability to thermal driving terms that vary over

| | |
|---|---|
| history of the two-phase flow. In Schmeling et al (2018) we studied this effect in detail and showed that choosing $\delta=d$ describes the thermal non-equilibrium only for intermediate term evolutions, not short period thermal variations (e.g. Fig. 8 in that paper). For $\delta<d$ the heat transfer coefficient $k$ will be larger than yours, so you probably overestimate thermal non-equilibrium for short term thermal variations. My recommendation: As it is quite common in literature to use the $\delta=d$ assumption for simplicity you should keep this assumption and address and discuss this point. | timescales that are "long" per your 2018 paper. Lines 176-184 |
| 3.) You don't say how you solve the equations. Please add a short section on the numerical method, grid resolution etc. | I added a section on my numerical solution methods. Section 2.3, line 220-227 |
| 4.) The Appendices D and E contain very interesting model results. In my opinion they should be moved to the main text. | Yes, both reviewers point out the problematic flow between the main text and the Appendices. I moved these sections into the main text to improve the flow. |
| 5) Discussion. In section i) you introduce the term "disequilibrium heating". This term should more rigorously be defined. In this section (e.g. Line 208) you estimate the heat budget due to disequilibrium heating by multiplying the excess infiltration temperature $\Delta T$ _by $k$ _to get a volumetric heat generation rate. According to eq. (2) you should use the disequilibrium temperature difference $T_f - T_s$ _rather than $\Delta T$,... | Agreed. I use $(T_f - T_s)_{max}$ = 2-5% of $\Delta T$ instead of $\Delta T$ and explain this in the context of the implied disequilibrium heat exchange. Lines 354-359

 I also define this term as I am using it in multiple places, lines 38, 211-214, 238-240, 352-355. |
| 6) Line 258, 260. Here you speculate about rheological weakening due to disequilibrium heating. Again, assuming 100 K as a possible temperature increase is a probably an overestimate given that the disequilibrium temperature difference $T_f - T_s$ _is one to two orders of magnitude smaller than $\Delta T$. And: I have checked the activation energies and volumes of Hirth and Kohlstedt (2003) and I don't get your factors of order 1/62. I get something like 1/20 at most for constant stress, and 1/3 for constant strain rate. Given the smaller temperature difference of order 10 K reduces this effect even more to a factor 1/1.3 or something like this, which is still worthwhile to mention. | Yes, fixed – I had used the E value for wet dislocation creep and the full DT. I have now revised this per your suggestion using $(T_f - T_s)_{max,}$ roughly 20% of $\Delta T$; lines 355-359

 Yes. |

**Minor points:**

| | |
|---|---|
| 7.) Line 308: you may note here that 1/z is the dimensionless channel velocity (but see also comment 13). | Yes, 1/z (now 1/ζ) is indeed the dimensionless channel velocity – see also response to #13 |

| | |
|---|---|
| 8.) Line 334. Are $\phi_{in}$ and $\phi_{out}$ identical to $\phi_f$ and $\phi_s$, respectively? Then you should use same symbols. | These are only identical if we take the 'end member' case where the channels are pure fluid and walls are solid. I was trying to say here that this does not need to be the case, but is now clarified (167-169) |
| 9.) Line 337. You choose $A$ and $\beta$ independently, but they are geometric parameters for spheres, tubes and spheres. Particularly $\beta$ is defined for solid spheres, cylinders and plates, while $A$ is defined for fluid tubes, etc. | Yes, you are correct. I do this to investigate how the estimates of K are affected by a range in A and $\beta$ – clarified in lines 173-175. |
| 10) Line 340 to 345 or section 2: Please specify the boundary conditions more rigorously, for both $T_s$ and $T_f$ at x = 0 and at the other side of the domain. You should clearly state that $T_s'$ is also raised to 1 while you increase the influx temperature of $T_f'$. | Yes, your point here and #13 below clearly show that the boundary and initial conditions need to be clarified. |
| 11.) Line 363. Delete "migration" | OK |
| 12.) Line 366 – 367 and line 143 – 149. The difference between the disequilibrium front velocity of Kuznetsov (1984) and your eq. 3 is puzzling and should be discussed. Is it due to different scaling? Although both, Kuznetsov's and your eq. 3 are given as dimensional equations? Or is it an effect of using perturbation theory versus full solution of the PDE's? Or is it a misprint in Kuznetsov? Anyway, how did you derive and justify eq. (3)? | I **think** that this arises because he is using an analytic (perturbative) approach whereas I am solving the full system. I will explain this better, but using the velocity estimate in Kuznetsov as a starting point (a first guess), I find (empirically) that Eqn 3 best describes the velocity in my models. I state this in revision (line 267-268), but I am afraid that I don't have a derivation. |
| 13.) Fig. A2c causes confusion. From the x-label or figure caption we have $x'_{front}=(1/z)t'$. (5) This implies that the disequilibrium front has the non-dimensional velocity 1/z. But the fluid velocity may be written as $v_{channel}=x_f t$ (6) where $x_f$ is the position of a fluid particle. If we substitute $x_f$ and $t$ using the non-dimensionalization rules one gets $v_{channel}=x_{f'} v_{channel} kst' k_f=x_{f'} t' v_{channel}z=v'_{channel} v_{channel}z$ (7) with $v'_{channel}$ as non-dimensional fluid velocity. After elimination of $v_{channel}$ from both sides we have $v'_{channel}=1/z$ (8) which is in contradiction to eq. (5). Can you help me (and potential readers)? | The confusion lies here: the initial conditions needed to be better described (currently in Appendix D): For t<0, there is material flowing in the channels, at $v_{channel}$, but the channels are at the same initial temperature as the walls, $T_0 = T_s = T_f$. At t=0, the temperature of the material entering the channels at x=0 is perturbed. So, what I mean by **$x_{front}$ in the old manuscript is not the location of the disequilibrium front**. Instead it is the location of the in-channel material that entered the domain at x=0 at t=0. This material moves at the speed $v_{channel}$ relative to the walls and at a later time t, it is located at x=$x_{front}$ = $v_{channel}$ t. Therefore, there is no contradiction, and 1/z is indeed the dimensionless velocity of material in the channels. I think the word "front" here is causing the confusion. In the revised paper, I changed this to $x_{pert}$ to more clearly indicate that this is the location of the material that entered with a perturbed temperature. Line 238-240. |
| 14.) Line 383. Sentence strange, probably delete one of the "is" or insert "which" | OK |

| | |
|---|---|
| 15.) Line 390 – 391. Which "blue lines"? Do you mean the dashed lines or the double arrows? | Thanks for catching a typo: these should refer to the double arrows (they used to be blue in a previous version) |
| 16.) Line 391: "wavelength" probably to be replaced by "period" | No, it is wavelength: I mean the peak-to-peak distance (at a fixed time) |
| 17.) Line 149. I don't see the strong function of $k$ _in Fig. A4. | Ok, it should refer to the functional dependence on $d$, which in turn strongly controls K; Figure 3 caption |
| 18.) Line 150 – 159. You clearly describe the exponential decay of disequilibrium. Could you elaborate a bit on the decay rate for the step function case? | The sentence is confusing… the exponential decay I am referring to is a spatial decay as a function of distance along the transport direction.  So, the successive peaks in the blue curve on figure A2b in the previous manuscript should go down by roughly the same factor as they are their distance apart is roughly the same.

In the revised manuscript, I show what I mean by a best-fit exponential decay – it is a spatial decay – used to estimate δ, the width of the TRZ (Figure 5 and its inset). |
| 19.) Line 163: delete one of the parantheses ")" in the first *tanh* term | Yes |
| 20.) Conclusion: Here I suggest to repeat the meaning of the abbreviations CLM, TRZ again | OK |

**Overall:**

"This paper investigates the potential for channelized melt transport into the base of the thermal lithosphere to supply an elevated localized heat flux. A simple modelling approach is used with a series of idealized forcing scenarios. The results include calculations of the scale of the thermal reworking zone and estimates of the overall heat supply. The modelling approach is heavily idealized and so is subject to significant limitations. The writing of the paper was hard to follow in parts. This could be improved by restructuring as a coherent whole without the back-and-forth use of appendices to develop both theory and results. However, the topic is interesting and the modelling is a useful starting point that makes a good contribution to analyzing the problem. Overall, I think the paper should be *accepted subject to minor revisions.*"

**Thank you very much for your review.  I appreciate the thoughtful and constructive comments you have provided; I gratefully acknowledge they will improve the paper and increase its impact and readability.**

Here is a brief summary of major changes made to the paper:
1. Calculations repeated with a system of equations including axial conduction terms – also a comparison of solutions with and without diffusion in the step-function perturbation case
2. Material in previously in Appendices is now within the flow of the main text.
3. Introduction and discussion of possible definitions of an effective Pe number – revision of parameter range for channel spacing *d* to ensure Pe > 10 here
4. Modifications to results:
    a. Reduction in the estimated contribution of disequilibrium heating as defined here to the LAB heat budget
    b. Scaling exponent for TRZ width δ as a function of characteristic perturbation timescale is between n=1 to 2, whereas it was closer to 2 without diffusion term
5. A short section describing numerical methods
6. Clarification of notations for heat transfer coefficient ($K$ not k; $K_s$ and $K_f$); dimensionless velocity is now ζ not $z$ (which seemed like a coordinate)

Below I address each of your comments and point to revisions (line numbers) in the revised manuscript (changes marked in red on revised PDF).

**General comments**

| 1. Explanation of model, especially relating to the heat transfer and channel spacing: In the specific comments section, I give several suggestions for how I thought the model and results could be explained more clearly. This includes some suggested revisions to the notation. | Thank you for this comment.  As R1 has pointed out also, the grounds for neglecting axial conduction are not valid for some of the models I consider. |
|---|---|
| | I have redone these calculations including the diffusion terms to test the robustness of my interpretations of the TRZ and the overall heat LAB budget for large Peclet numbers.  The conclusions of the paper have not undergone significant modification and the overall findings of the previous version of this study hold. In summary, key results that remain unchanged are: |
| | 1. A limit is set on the importance of disequilibrium heat exchange within the lowermost continental lithosphere, but the heat budget is now revised to be |

| | lower than previously, comparable to that estimated from the deposition of latent heat (lines 354-360; 374-377).

2. Documenting the rate at which the zone of disequilibrium heat exchange progresses inward into the domain from the inlet.

3. Showing the likelihood that a TRZ forms at the base of the CLM for geologically-reasonable parameters and that the width of the TRZ is proportional to the characteristic thermal perturbation timescale. |
|---|---|
| My main concern
in this area relates to the modelling of the heat transfer process. Physically, I would think that macroscale heat transfer results from microscale diffusion (i.e. thermal conduction), but the paper states that axial conduction is neglected. But looking at the appendix (C2), k is proportional to an effective thermal conductivity divided by the square of some length scale (d in the equation). The correct choice/s of length scale is the crucial issue (the square is clear from dimensional grounds). The authors say that d is the channel spacing, but elsewhere (L316) say that d is the particle diameter. These are obviously very different. So the relevant length scale needs much better justification and the role of conduction (equivalently diffusion) in the model should be clearer. | I have clarified the confusion of using d for both channel spacing and particle diameter (now $p$; lines 149-50) and axial conduction is now included.

The relative importance of diffusion and disequilibrium heat exchange is now addressed in the broadening observed for a step function lines 252-260. |
| 2. Simplifications in modelling approach: There are numerous simplifications inherent in the modelling approach. These are generally mentioned in the text but I felt the paper would benefit from more analysis of the relative importance of the various simplifications made. Two related simplifications that seem especially important to me relate to the parameters φ (fraction occupied by channel), the make up of the channels, and thermal (and perhaps chemical feedbacks).

It is not entirely clear
whether the channels are envisaged as purely liquid, narrow dikes surrounded by entirely solid rock or much wider bodies of partially molten rock, where a channel is distinguished as having a higher melt fraction. In either case, it is clear that the properties of these channels are in practice determined that the operative dynamics and it is a large simplification to just impose them. There must also | Yes, related to 1 above, I discuss the relative importance of axial conduction terms in the equations and address how including these affects the overall story of the paper. This is done in (new) Figure 3c&d and Lines 252-260.

Chemical exchange is ignored here entirely and this is indeed a limitation, but it will be more clearly stated in the revised paper. Lines 85-86.

As I mentioned in lines 330-339 of the previous manuscript, the channels may be a high-porosity region within a lower-porosity surrounding region. To explore the 'end-member' upper limit to the disequilibrium heat exchange, I consider the case where the channels are purely liquid and walls are solid. This is now further clarified in the main text and not in an appendix (see response to #3 below) – Lines 167-169 |

| | |
|---|---|
| be feedbacks between any thermal reworking process and the channels themselves but this can't be investigated within this type of model, as the channel properties are just imposed. | Yes, transport and channelization feedbacks cannot be explored in this limited approach.  This is stated in lines 333-335 |
| 3. Paper structure: Significant aspects of the paper were hard to follow. I was less concerned about appendix A (but also don't see why a few short paragraphs couldn't be included in the introduction). Appendix C develops substantial aspects of the model (including aspects novel or specific to this study) to such an extent that the description in the main text relies heavily on material in the appendix (e.g. the discussion of k, $k_f$ and $k_s$, which are crucial to the paper). Appendix B is rather more technical, but the meaning of symbols developed there is relied on elsewhere. So it should eitherbe incorporated into the main text, or care should be taken such that all notation is properly defined in the main text at least.

Appendix D and especially appendix E, given that it is perhaps the most 'realistic' scenario considered, also belong in the main results section. The summary given relies on notation developed in the appendices as well as figures only reported in the appendices. For this style of journal, the back-and-forth between main text and appendices is hard to justify. | OK, thank you for this comment.  I feel your suggestions will greatly improve the flow of the paper and strengthen its impact.  This is also in line with R1's comments on the organization of the Appendices and the material in the text.

Material previously in Appendices have been incorporated into the main text and into the flow/story of the paper.

Yes. |

Technical comments:

| | |
|---|---|
| 4. L33–45 or final paragraph of introduction: Consider referring to body of work relating to thinning of the thermal lithosphere in arc settings (e.g. England and Katz, 2010, https://doi.org/ 10.1038/nature09417, Perrin et al., 2016, https://agupubs.onlinelibrary.wiley.com/doi/10. 1002/2016GC006527 and Rees Jones et al., 2018, https://doi.org/10.1016/j.epsl.2017.10.015.) | Yes, I now include some of these citations (lines 27-30; 59-60). I now refer to this when considering other contributions to the heat budget at the LAB. Lines 374-377 |
| 5. L51-54: this is a very significant simplification as it precludes any feedbacks between the channels and the process(es) that create them. | Yes. It is highlighted in the beginning of the Discussion; lines 325-335. |
| 6. L58: 'v is transport velocity' needs a bit more explanation (transport velocity of what?). Also I assume from the equations that the solid is not moving but this could be stated more clearly in the text. I don't really understand why you introduce a new symbol $v_{channel}$ when it seems to be the same as v. The cartoon sketch in figure 1 is also a bit unclear as to whether v is the fluid velocity within the narrow channels in the zoomed in circles or some kind of average? | OK, yes, this is a typo. The velocity is now written exclusively as $v_{channel}$, the transport velocity of material inside the channels. I retain the subscript '*channel*' to specify that this is the average rate of relative motion of material within and outside channels. Also, the physical meaning of $v_{channel}$ is now clarified earlier in Figure 1 ("average velocity= $v_{channel}$") and in text (lines 112-113 and elsewhere). |
| 7. Eqs. 1–2: This way of defining $k_f$ and $k_s$ could be clearer. The notation is also potentially confusing as k has different units from $k_s$ and $k_f$. Suggest changing one of the symbols. | OK, your point about the units being different is very good and I have changed the notation so *K* is only used as the effective heat transfer coefficient. (I avoided the lowercase *k* to avoid confusion with the common symbol for thermal conductivity.) |
| 8. Figure 1: These time-dependent forcings have very different total energy inputs which could be emphasized a bit more, perhaps. | OK this is a good point, and I touch on this in the discussion of how the TRZ width scales with the characteristic perturbation timescale– Lines 317-325. |
| 9. L87 & L109: 'across channel walls' sounded a bit strange because the fluid flow seemed to be vertical so there wouldn't be much flow across channel walls, since the walls in the sketch are also near vertical. | Clarified; I mean relative motion between material inside and outside channels. Lines 112-113 |
| 10. L104-112: consider phrasing this discussion in terms of a Peclet number. | Yes, this was also brought up by R1 and I have included this in the revision. Lines 205-220 |
| 11. L136: Think 'duration' was intended rather than 'amplitude.' | Yes, you are correct. |
| 12. L138–: Think that this section would be easier to understand if text from appendix (and especially figures) was included in the main text. | OK, agreed. |

| | |
|---|---|
| 13. Figure 2: This is a useful figure. But I think plots against x at a series of t values are also useful complementary way to show the same data. | Yes, I have now included both views, T-x and T-t in a composite figure 6 -- see c and d |
| 14. L174: 10 m. | Yes, I need the units |
| 15. Figure 3: Consider plotting agains the theoretical scaling to collapse all the data on a single line. | I thought about this, but decided against it because the effect of $d$ on δ is important to show visually. However this figure is now better annotated and visually clarified. |
| 16. Figure 4 & L231: I wondered if this velocity range was rather low, for example when compared to typical asthenospheric melt velocities which might be an order of magnitude larger. | Although I only consider one representative $v_{channel}$ = 1 m/yr, I mention that increasing $v_{channel}$ will increase both $V_{diseqm}$ and δ since, for fixed channel geometry (φ, $d$), both depend linearly on $v_{channel.}$ Line 380-383; 392-393. |
| 17. L203–224: Perhaps it would make more sense to consider the overall LAB heat budget rather than one component. | Agreed, however the stated goals of this work are to place limits on this one process, namely disequilibrium heat exchange. I state this explicitly here again, but place this in the context of other work that examines advective heat transport and latent heat transport. Line 374-377. |
| 18. L305: z is an odd choice of symbol (looks more like a vertical coordinate) and could be defined more clearly. | OK. This is in keeping with some of the previous literature I cite (e.g., Spiga and Spiga and Kuznetsov). I can see how this would be confusing though and have now clarified it. I use ζ instead of z throughout. Also, because ζ is not as conceptually simple as the channel volume fraction φ, where possible I avoid referring to ζ values but instead refer to φ values as there is a 1:1 mapping between them; 237-245. |
| 19. L310: Might benefit from a brief discussion of the numerical methods used. | Yes, agreed. I plan to add a short section on this. Lines 220-227 |
| 20. L311 & 316: d appears to be used for two different quantities | Yes, this is corrected by using a different symbol for particle diameter, $p$. Line 149-150 |
| 21. eqs. C1 & C2: check whether the minus sign is correct. This looks like it should be related to the harmonic mean of two conductivities (it would be with a plus sign). And the equations would be problematic if the term in square brackets were zero. | Yes, absolutely; As also pointed out by R1, another typo. |
| 22. L325: Not sure where this range came from originally but I don't think it would be appropriate if | OK. I agree that it is tricky in a 'coarse grained' model such as this to connect to microscopic geometry. My intention here is to illustrate what reasonable |

| | |
|---|---|
| the model is intended to be of a porous flow, it sounds more like a pipe flow argument. | numbers might be for A and β... as Reviewer 1 suggested, I can connect to some previous work to motivate this better.  Line 173-184 |
| 23. Figure A3: Could benefit from better formatting to match the standard of the other figures | OK – I am guessing you mean panels (c) and (d) in particular. I have now combined this into a figure illustrating both the T-x and the T-t view of the model solutions (Figure 6). |

---

## Referee Report (RR1)

**Review of the revised version of "Assessing the role of thermal disequilibrium in the evolution of the lithosphere-asthenosphere boundary: An idealized model of heatexchange during channelized melt-transport"**

Harro Schmeling

June 30, 2022

A revised version has been submitted in which all points and concerns of my previous review have been properly addressed. The manuscript has improved significantly. Very nicely the effect of axial diffusion is included and discussed now, and the discussion about the energy budget has been improved. I detected only one point which would confuse the reader:

Your definition of Peclet numbers (eq. $8 - 10$) is strange. You use a length scale $l$ defined as the product of an advection velocity and a diffusion time $t_f$ of channel width. Usually the Peclet number should give the ratio of an advective length scale divided by a diffusive length scale (both per time). Writing the characteristic diffusion time across a channel as $t_f = d_{channel}^2/\kappa_f$ , your definition reads as

$$Pe_1 = \frac{v_{channel} l}{\kappa_f} = \frac{v_{channel}^2 t_f}{\kappa_f} = \frac{v_{channel}^2 d_{channel}^2}{\kappa_f^2} = Pe_{channel}^2$$

where $\kappa_f$ is the fluid diffusivity, $d_{channel}$ is the channel width and $Pe_{channel}$ is a Peclet number based on advection through the channel and diffusion across the channel. The same argument applies to your second Peclet number which seems to be the square of a Peclet number base on advection along the channels and diffusion across the grains (solid). Thus your definitions are no real Peclet numbers sensu stricto. I suggest to use alternative definitions (such as $Pe_{channel}$ above, see also definitions in my previous review) even though they may be smaller than the ones you used in the manuscript now.

Line 222: Missing ")"

---

## Referee Report (RR2)

**Review of Revision 1 of "Assessing the role of thermal disequilibrium in the evolution of the lithosphere-asthenosphere boundary: An idealized model of heat exchange during channelized melt-transport"**

**I. GENERAL COMMENTS**

I thank the author for the detailed responses to the reviewers' comments. I think that the revised version of the manuscript has been improved in important respects. The structure of the paper is now much better. However, I still think the physical description of heat transfer could be improved (point 1) and think the expression of the scaling behaviour of the TRZ width, which was added in revision, needs reworking (point 2). Overall, given the strengths of the paper pointed out in our earlier reviews, I think the paper should be *accepted subject to minor/technical revisions.*

**II. SPECIFIC COMMENTS**

[In all the following comments, I refer to line numbers in the tracked changes version of the revised manuscript.]

1. **Explanation of model, especially relating to the heat transfer and channel spacing:** I want to return to this issue as I think could be more clearly described. In the full physical system, heat will be transferred by advection and diffusion (and latent heat, if there were phase change). The diffusion will occur both in the vertical (along channel) and horizontal directions (across channel, from the hot channel into the colder surrounding rock). In revision, an extra term was added to equations (1) and (2), representing vertical diffusion. This is absolutely fine. However, generally you expect horizontal diffusion to be much more important than vertical diffusion (because the horizontal length scale is smaller than the vertical scale). In a 1D (vertical) model, you cannot represent horizontal diffusion explicitly. Instead, in this type of model, the effect of horizontal diffusion is represented by the heat transfer term involving $K$.

   I think that the text of the paper should make it much clearer that this term involving $K$ arises from horizontal diffusion. At present, the first paragraph explaining $K$ (starting L132) emphasizes that $K$ is a proxy for the geometry of the channel wall interface. The block of text added starting on L173 starts to address the crucial issues. Based on dimensional and physical arguments, you would expect

that the timescale of heat transfer to be proportional to the square of a boundary layer dimension (since thermal diffusivity has units length$^2$/time). Then you assume (on the grounds explained around L180), that the boundary layer dimension is proportional to $d$, which gives you essentially equation (4). So I would recommend rewriting L132–183 to start with the essential physics (horizontal diffusion) and assumptions first, before moving on to the details are the channel geometry. I would try to limit switching between $K$, $K_{f,s}$ (which have different units to $K$, something I found confusing at first, and would ideally be avoided), and $C_{eff}$ as far as possible. L119 was also quite confusing in that it talked about diffusion being ignored and used the symbols $D_{f,s}$ which don't seem to appear elsewhere.

2. **Thermal reworking zone width scaling:** the expression given on L14, added in revision, is dimensionally inconsistent. The RHS doesn't have the same units as the LHS. I do not think the final result should be expressed with a term like $(\tau/d)^n$, given that $\tau/d$ is not dimensionless. Ideally, you want expressions like equation (14), where a dimensionless quantity is raised to some power. In the main results section, around L318, you have $\delta \sim \tau^n$, which doesn't have any dependence on $d$. But then the abstract (L14) and conclusion (L397) give a proportionality $d^{-n}$. It would be good to explain the dependence on $d$ and to try to write the final result in dimensionally consistent groupings. (A theoretical justification for the $n = 1$ scaling would be very good if possible, but I appreciate this might potentially not be straightforward.)

**III. TECHNICAL CORRECTIONS**

3. **L14:** $n \approx 2$ (remove word 'is').

4. **around L86:** I would say that a key limitation of this type of study is that the channel properties are imposed, rather than emergent dynamically. This simplification is well described later in the paper, but probably should be mentioned somewhere in the introduction more explicitly.

5. **L110:** perhaps should define $x$ (particularly as $x$ is often used a horizontal coordinate) and $t$.

6. **L202:** perhaps avoid extra spaces in, e.g., $T'_f$.

7. **L207:** I'm not entirely sure what $\zeta$ means (in particular, it is not entirely clear how it has been defined. Perhaps give a formula. I also think the choice of notation is a bit unusual.

8. **L281:** missing space before 'years.' (I noticed some other examples of this too.)

9. **Table 1:** missing link to appendix.

---

## Author Response (AR2)

Response to comments on revision 1                                     27 July, 2022
Mousumi Roy

I am very grateful to both Referees for the time taken to review this work and for their thoughtful and detailed comments–I gratefully acknowledge this in L454. All line numbers here and below refer to the track-changes version of the revised manuscript, where changes due to the responses below are highlighted in red.

I have also searched for and removed any remaining typos.

**Referee 1**

| | |
|---|---|
| 1. Your definition of Peclet numbers (eq. $8 - 10$) is strange. You use a length scale $l$ _defined as the product of an advection velocity and a diffusion time $t_f$ _of channel width. Usually the Peclet number should give the ratio of an advective length scale divided by a diffusive length scale (both per time). Writing the characteristic diffusion time across a channel as $t_f = d_{channel}^2/\kappa_f$, your definition reads as $$Pe_1 = \frac{v_{channel} l}{\kappa_f} = \frac{v_{channel}^2 t_f}{\kappa_f} = \frac{v_{channel}^2 d_{channel}^2}{\kappa_f^2} = Pe_{channel}^2$$ where $\kappa_f$ _is the fluid diffusivity, $d_{channel}$ _is the channel width and $Pe_{channel}$ _is a Peclet number based on advection through the channel and diffusion across the channel.

 The same argument applies to your second Peclet number which seems to be the square of a Peclet number base on advection along the channels and diffusion across the grains (solid). Thus your definitions are no real Peclet numbers sensu stricto.

 I suggest to use alternative definitions (such as $Pe_{channel}$ _above, see also definitions in my previous review) even though they may be smaller than the ones you used in the manuscript now. | Yes, I acknowledge that what I have is not strictly a Péclet number for the channel (your $Pe_{channel}$).

 Since the fluid diffusivity (your $\kappa_f$, my $\lambda_f/c_f$ in Eqns 1 and 2) is a parameter governs conduction within the channel material, it is proper to define $Pe_{channel}$, as you have, for the axial (along-transport direction).

 However, what I really want is to represent the key role played in the model by the heat transfer between channels and surroundings, namely the heat transfer that is represented by $K$, the heat transfer coefficient (the across-channel heat transfer). This is why I used a length scale given by velocity/$K_f$ (or $K_s$). This is also why I always refer to my Péclet number definitions as "effective" Péclet numbers (L XX).

 Importantly, this allows me to involve a combination of material parameters, from both within and outside the channels.

 I have decided to keep the definitions as they are, but I now explicitly state my thinking in using such a definition and I also acknowledge your argument that these are not the same as $Pe_{channel}$.

 This way, the reader is made aware of the unconventional definition and the ideas behind it.  Lines 228-233. |
| 2. Line 222: Missing ")" | Thank you for catching this – fixed. |

**Referee 2**

| | |
|---|---|
| 1. Explanation of model, especially relating to the heat transfer and channel spacing: I want to return to this issue as I think could be more clearly described. In the full physical system, heat will be transferred by advection and diffusion (and latent heat, if there were phase change). The diffusion will occur both in the vertical (along channel) and horizontal directions (across channel, from the hot channel into the colder surrounding rock). In revision, an extra term was added to equations (1) and (2), representing vertical diffusion. This is absolutely fine. However, generally you expect horizontal diffusion to be much more important than vertical diffusion (because the horizontal length scale is smaller than the vertical scale). In a 1D (vertical) model, you cannot represent horizontal diffusion explicitly. Instead, in this type of model, the effect of horizontal diffusion is represented by the heat transfer term involving K.

I think that the text of the paper should make it much clearer that this term involving K arises from horizontal diffusion. At present, the first paragraph explaining K (starting L132) emphasizes that K is a proxy for the geometry of the channel wall interface. The block of text added starting on L173 starts to address the crucial issues. Based on dimensional and physical arguments, you would expect that the timescale of heat transfer to be proportional to the square of a boundary layer dimension (since thermal diffusivity has units length$_2$/time). Then you assume (on the grounds explained around L180), that the boundary layer dimension is proportional to d, which gives you essentially equation (4). So I would recommend rewriting L132–183 to start with the essential physics (horizontal diffusion) and assumptions first, before moving on to the details are the channel geometry. I would try to limit switching between K, $K_{f,s}$ (which have different units to K, something I found confusing at first, and would ideally be avoided), and $C_{eff}$ as far as possible.

L119 was also quite confusing in that it talked about diffusion being ignored and used the symbols $D_{f,s}$ which don't seem to appear elsewhere. | You raise a very good point – it is something I struggled with, since the heat transfer coefficient and its meaning are discussed below the first, general, sketch of the model.

I have now followed your suggestion and discussed this idea of a heat transfer coefficient that is a proxy for the horizontal diffusion you mention. To be general, however, I refer to it as diffusion perpendicular to the transport direction in this 1D model.  L133-152.

Yes, this is a good point – it was confusing as $D_f$ and $D_s$ are defined later... I have removed the reference to these, but explained what I mean by diffusion terms, in L120-121 |
| 2. Thermal reworking zone width scaling: the expression given on L14, added in revision, is dimensionally inconsistent. The RHS doesn't have the same units as the LHS. I do not think the final result should be expressed with a term like $(\tau/d)_n$, given that $\tau/d$ is not dimensionless.

Ideally, you want expressions like equation (14), where a dimensionless quantity is raised to some power. In the main results section, around L318, you have $\delta \sim \tau_n$, which doesn't have any dependence on d. But then the abstract (L14) and conclusion (L397) give a proportionality $d_{-n}$. It would be good | The idea of mentioning how the width $\delta$ of the TRZ depends on $\tau$ (timescale) and $d$ (channel spacing) as a scaling relation is in the same spirit as saying "the mass of an object scales with a linear dimension, $r^{3}$". Or, to use an example from biology, we might say that the "metabolic rate scales as (mass)$^{\beta}$". There is an implied prefactor that makes the corresponding equation dimensionally correct. Therefore, I don't agree that all scaling should only be expressed in dimensionless groupings. |

| to explain the dependence on d and to try to write the final result in dimensionally consistent groupings. | I do agree, however, that this may be confusing to some readers. I have now explicitly stated what I mean: that $\delta$ is found to be "proportional to" the various combination of parameters; L14, 315-316, 333, 412. |
|---|---|

*TECHNICAL CORRECTIONS*

| 3. L14: n ≈ 2 (remove word 'is'). | Thanks for catching this – done. |
|---|---|
| 4. around L86: I would say that a key limitation of this type of study is that the channel properties are imposed, rather than emergent dynamically. This simplification is well described later in the paper, but probably should be mentioned somewhere in the introduction more explicitly. | This is a good point and is now spelled out in L 70-75 |
| 5. L110: perhaps should define x (particularly as x is often used a horizontal coordinate) and t. | OK – now added; L 115 |
| 6. L202: perhaps avoid extra spaces in, e.g., T′f. | Yes, thank you for catching this typo – now fixed, throughout. |
| 7. L207: I'm not entirely sure what $\zeta$ means (in particular, it is not entirely clear how it has been defined. Perhaps give a formula. I also think the choice of notation is a bit unusual. | The quantity $\zeta$ is a weighted heat capacitance ratio (its definition and formula are given on line 209-210). The notation was modified after comments by R2 on the first version of this paper – it used to be $z$, which was deemed inappropriate as resembled a vertical coordinate. |
| 8. L281: missing space before 'years.' (I noticed some other examples of this too.) | Fixed. |
| 9. Table 1: missing link to appendix. | Fixed; it now links to the section on the heat transfer coefficicent, 2.1 |